

# A study of QCD radiation in VBF Higgs production with Vincia and Pythia

Stefan Höche[1], Stephen Mrenna[1], Shay Payne[2],
Christian T Preuss[2*] and Peter Skands[2]

**1** Fermi National Accelerator Laboratory, Batavia, IL, 60510, USA
**2** School of Physics and Astronomy, Monash University, Wellington Road, Clayton, VIC-3800, Australia

⋆ christian.preuss@monash.edu

## Abstract

We discuss and illustrate the properties of several parton-shower algorithms available in Pythia and Vincia, in the context of Higgs production via vector boson fusion (VBF). In particular, the distinctive colour topology of VBF processes allows to define observables sensitive to the coherent radiation pattern of additional jets. We study a set of such observables, using the Vincia sector-antenna shower as our main reference, and contrast it to Pythia's transverse-momentum-ordered DGLAP shower as well as Pythia's dipole-improved shower. We then investigate the robustness of these predictions as successive levels of higher-order perturbative matrix elements are incorporated, including next-to-leading-order matched and tree-level merged calculations, using Powheg Box and Sherpa respectively to generate the hard events.



# 1 Introduction

Higgs boson production via Vector Boson Fusion (VBF) — fig. 1 — is among the most important channels for Higgs studies at the Large Hadron Collider (LHC). With a Standard-Model (SM) cross section of a few pb at LHC energies, VBF accounts for order 10% of the total LHC Higgs production rate [1]. The modest rate is compensated for by the signature feature of VBF processes: two highly energetic jets generated by the scattered quarks, in the forward and backward regions of the detector respectively, which can be tagged experimentally and used to significantly reduce background rates. Moreover, the distinct colour flow of the VBF process at leading order (LO), highlighted by the coloured thick dashed lines in fig. 1, strongly suppresses any coherent bremsstrahlung into the central region, leaving this region comparatively clean and well suited for precision studies of the Higgs boson decay products. With over half a million Higgs bosons produced in the VBF channel in total during Run II of the LHC and a projection that this will more than double during Run III, studies of this process have already well and truly entered the realm of precision physics.

On the theory side, the current state of the art for the $H+2j$ process in fixed-order perturbation theory is inclusive next-to-next-to-next-to-leading order QCD [2], fully differential next-to-next-to-leading order (NNLO) QCD [3–6] and next-to-leading-order (NLO) electroweak (EW) calculations [7]. These calculations of course only offer their full precision for observables that are non-zero already at the Born level, such as the total cross section and differential

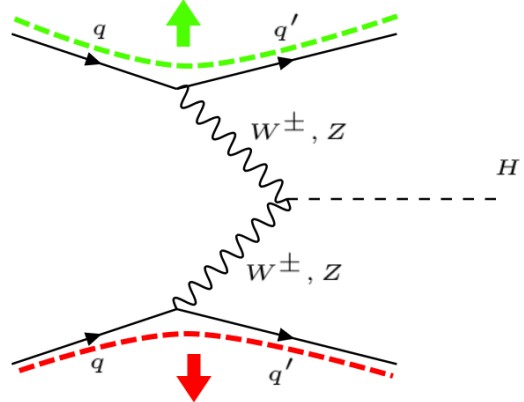

Figure 1: QCD colour flow of the LO VBF Higgs production process. Due to the kinematics of the interaction, QCD radiation is directed in the forward region of the detector.

distributions of the Higgs boson and tagging jets. For more exclusive event properties, such as bremsstrahlung and hadronisation corrections, the most detailed description is offered by combinations of fixed-order and parton-shower calculations. To this end, two recent phenomelogical studies [8,9] compared different NLO+PS simulations among each other as well as to NLO and NNLO calculations. These comparative studies catered to two needs; firstly, the reliability of matched calculations was tested in regions where resummation effects are small. Furthermore, a more realistic estimate of parton-shower as well as matching uncertainties was obtained by means of different shower and matching methods in independent implementations.

The earlier of the two studies [8] highlighted that different NLO+PS implementations describe the intrinsically coherent radiation in this process quite differently, and that the uncertainties arising from the choice of the shower and matching implementation can persist even at the NLO-matched level. Among its central results, the study [8] confirmed the observation of [10] that PYTHIA's default shower [11–13] describes the emission pattern of the third jet poorly, essentially missing the coherence of the initial-final dipoles. This effect was most pronounced for MADGRAPH_AMC@NLO [14] + PYTHIA, for which a global recoil scheme must be used in both the time-like and space-like shower in order to match the subtraction terms implemented in MADGRAPH_AMC@NLO. For POWHEG-BOX [15] + PYTHIA, the difference persisted when using the global recoil scheme[1]. However, changing to PYTHIA's alternative dipole-recoil scheme [16], which should reproduce coherence effects more faithfully, improved the agreement, both with calculations starting from $H+3j$ as well as with the angular-ordered coherent shower algorithm in HERWIG 7 [17].

The more recent study [9] highlighted a number of interesting aspects of vector boson fusion that can be exploited to enhance the signal-to-background ratio in future measurements: Firstly, if the Higgs boson is boosted, the $t$-channel structure of the VBF matrix elements leads to less QCD radiation when compared to the irreducible background from gluon-gluon fusion. Secondly, it was found that a global jet veto provides a similarly effective cut as a central jet veto, leading to much reduced theoretical uncertainties, and in particular eliminating the need to resum non-global logarithms associated with inhibited radiation in the rapidity gap. Despite a good overall agreement between fixed-order NNLO and NLO-matched parton shower predictions, the study also pointed out a few subtle disagreements for highly boosted Higgs boson topologies. In these scenarios, the standard fixed-order paradigm of operating with a single factorisation scale is no longer appropriate, because higher-order corrections should be resummed individually for the two impact factors in the structure-function approach.

The uncertainties arising from matching systematics in vector-boson-fusion and vector-boson-scattering processes (VBS) have also been studied in the past [18] with rather good agreement between different showers at the level of $H+3j$ NLO+PS calculations [19], although in that study, only the POWHEG matching scheme was considered. Very recently, two extensive reviews [20,21] collected experimental results and theoretical developments in VBS processes in view of the high-luminosity upgrade of the LHC as well as future colliders. A summary of Monte Carlo event generators used in the modelling of VBS processes in ATLAS was presented in [22].

On the experimental side, recent studies of VBF Higgs production by ATLAS [23,24] and CMS [25,26] have used PYTHIA's default shower algorithm matched to the NLO via the POWHEG technique, with only one of them [23] employing PYTHIA's dipole option. The associated modelling uncertainties, and ways to reduce them, therefore remain of high current relevance.

We extend the comparative study of [8] to include the new VINCIA sector-antenna shower [27] that has become available starting from PYTHIA version 8.304. Based on find-

---

[1]We note that the global recoil scheme is the default choice only for PYTHIA's space-like DGLAP shower, while the time-like DGLAP shower uses a dipole-like recoil scheme per default.

ings pertaining to antenna [28–31] and dipole [32–35] showers, we expect that, at least at leading colour, VINCIA's showers capture QCD coherence effects in VBF more accurately than PYTHIA's default shower. To this end, we note that the emitter-recoiler agnostic antenna recoil employed in VINCIA is free of adverse kinematic effects [36]. We also consider two new observables designed to further probe the amount of radiation by measuring the summed transverse energy $H_T$ for $|\eta| < 0.5$ and for $|\eta - \eta_0| < 0.5$ respectively, where $\eta_0$ is the midpoint between the two tagging jets. To investigate the robustness of the predictions, we include not only POWHEG-BOX + PYTHIA [13, 15] but also a new dedicated implementation of the CKKW-L merging scheme [37–39] for sector showers [40], with hard events with up to four additional jets generated by SHERPA 2 [41, 42]. We emphasise that this is currently the only multi-jet merging approach in PYTHIA 8.3 which can handle VBF processes[2]. Additionally, we highlight the systematic uncertainties arising from the use of vetoed showers in the POWHEG scheme and make recommendations for settings related to the use of these in PYTHIA.

This study is structured as follows. We begin with an overview of the setup for our simulations in section 2; starting with an overview of the fixed-order, shower, matched and merged calculations and leading towards a description of the analysis we perform. We then move on to discuss the results of our analysis in section 3, with our conclusions and recommendations listed in section 4.

## 2 Setup of the Simulation

We consider Higgs production via VBF in proton-proton collisions at the high-luminosity LHC with a centre-of-mass energy of $\sqrt{s} = 14$ TeV.

The simulation is factorised into the generation of the hard process using SHERPA 2 (for the LO merging samples) and POWHEG-BOX v2 (for the NLO matched samples) and subsequent showering with PYTHIA 8.306. A cross check is also performed using PYTHIA's internal Born-level VBF process. Details on the hard-process setups are given in section 2.1.

Since we expect the VINCIA antenna shower to account for coherence more faithfully than does PYTHIA's default "simple" $p_\perp$-ordered DGLAP shower, we take VINCIA's description as the baseline for our comparisons, contrasting it to PYTHIA's default and "dipole-recoil" options. Details on the shower setups are given in section 2.2.

Higher fixed-order corrections are taken into account at NLO+PS accuracy via the POWHEG scheme, and for VINCIA also in the CKKW-L scheme up to $\mathcal{O}(\alpha_s^4)$. We expect that these corrections will be smaller for coherent shower algorithms than for incoherent ones, hence these comparisons serve both to test the reliability of the baseline showers and to illustrate any ambiguities that remain after these corrections are included. Details on the matching and merging setups are given in section 2.3.

Finally, in section 2.4, we define the observables and the VBF analysis cuts that are used for the numerical studies in section 3.

Note that, since we are primarily interested in exploring the coherence properties of the perturbative stages of the event simulation, most of the results will be at the so-called "parton level", i.e. without accounting for non-perturbative or non-factorisable effects such as hadronisation, primordial $k_T$, or multi-parton interactions (MPI). Although this is not directly comparable to physical measurements (nor is the definition universal since different shower algorithms define the cutoff differently), the factorised nature of the infrared and collinear safe observables we consider imply that, while non-perturbative effects may act to smear out the perturbative differences and uncertainties, they would not in general be able to obviate them,

---

[2]We do note that a technical (but due to the use of incoherent IF kinematics unphysical) fix was introduced in PYTHIA 8.242 and is planned to be re-implemented in a future version of PYTHIA 8.3.

thus making studies of the perturbative stages interesting in their own right. Nevertheless, with jet $p_\text{T}$ values going down to 25 GeV and $H_\text{T}$ being sensitive to the overall amount of energy scattered into the central region, we include further comparisons illustrating the effect of non-perturbative corrections at the end of section 3.

## 2.1 Hard Process

For the parton-level event generation, we use a stable Higgs boson with a mass of $M_\text{H} = 125$ GeV, and we set the electroweak boson masses and widths to

$$M_Z = 91.1876 \text{ GeV}, \quad \Gamma_Z = 2.4952 \text{ GeV}, \tag{1}$$
$$M_W = 80.385 \text{ GeV}, \quad \Gamma_W = 2.085 \text{ GeV}.$$

Electroweak parameters are derived from this set with the additional input of the electromagnetic coupling constant at the $Z$ pole ($\alpha(M_Z)$ scheme, EW_SCHEME = 2 in SHERPA):

$$\frac{1}{\alpha(M_Z)} = 128.802. \tag{2}$$

We treat all flavours including the bottom quark as massless and use a diagonal CKM mixing matrix. In both SHERPA and POWHEG-BOX, we use the CT14_NNLO_as118 [43] PDF set provided by LHAPDF6 [44] with the corresponding value of $\alpha_\text{S}$. For the sample generated with PYTHIA's internal VBF implementation, we use its default NNPDF23_lo_as_0130_qed PDF set [45,46].

We consider only VBF topologies, neglecting Higgsstrahlung contributions which appear at the same order in the strong and electroweak coupling. Identical-flavour interference effects are neglected in events generated with POWHEG-BOX and PYTHIA, but are included in events obtained with SHERPA, although their impact was found to be small [9]. At NLO, the process is calculated in the structure function approximation, neglecting interferences between the two quark lines. For both, internal and external events, only a single scale will be assigned per event, notwithstanding that different scales could in principle be assigned to the two forward-scattered quarks. Differences pertaining to the scale assignment in internal and external events will be discussed in section 3.1.

Tree-level event samples with up to four additional jets are generated using an HPC-enabled variant of SHERPA 2 [41,42], utilising the COMIX matrix-element generator [47]. To facilitate efficient parallelised event generation and further processing, events are stored in the binary HDF5 data format [42]. The factorisation and renormalisation scales are chosen to be

$$\mu_\text{F}^2 = \mu_\text{R}^2 = \frac{\hat{H}_\text{T}^2}{4} \quad , \text{with} \quad \hat{H}_\text{T} = \sum_j p_{\text{T},j} + \sqrt{M_\text{H}^2 + p_{\text{T,H}}^2}, \tag{3}$$

and jets are defined according to the $k_\text{T}$ clustering algorithm with $R = 0.4$ and a cut at 20 GeV.

PYTHIA's internal events are generated with scales governed by the two switches SigmaProcess:factorScale3VV and SigmaProcess:renormScale3VV, respectively. Their default values = 2 and = 3, respectively, correspond to the choices

$$\mu_\text{F}^2 = \sqrt{m_{\text{T},V_1}^2 m_{\text{T},V_2}^2} \equiv \sqrt{(M_{V_1}^2 + p_{\text{T},q_1}^2)(M_{V_2}^2 + p_{\text{T},q_2}^2)}, \tag{4}$$

$$\mu_\text{R}^2 = \sqrt{m_{\text{T},V_1}^2 m_{\text{T},V_2}^2 m_{\text{T,H}}^2} \equiv \sqrt[3]{(M_{V_1}^2 + p_{\text{T},q_1}^2)(M_{V_2}^2 + p_{\text{T},q_2}^2)m_{\text{T,H}}^2}, \tag{5}$$

with the pole masses of the exchanged vector bosons $M_{V_1}$, $M_{V_2}$, the transverse mass of the Higgs boson $m_{\text{T},H}$, and the transverse momenta of the two final-state quarks $p_{\text{T},q_1}$, $p_{\text{T},q_2}$.

For NLO calculations matched to parton showers, we consider the POWHEG [48, 49] formalism. POWHEG samples are generated with POWHEG-BOX v2 [15, 50] with the factorisation and renormalisation scales chosen as

$$\mu_F^2 = \mu_R^2 = \frac{M_H}{2}\sqrt{\left(\frac{M_H}{2}\right)^2 + p_{T,H}^2}. \tag{6}$$

Since the study in [8] did not find any significant effect from the choice of the "hdamp" parameter in POWHEG, we do not include any such damping here, corresponding to a choice of hdamp = 1.

## 2.2 Showers

The hard events defined above are showered with the three following shower algorithms, which are all available in PYTHIA 8.306:

- VINCIA's sector antenna shower [27]. The "sector" mode is the default option for VINCIA since PYTHIA 8.304 and also enables us to make use of VINCIA's efficient CKKW-L merging [40]. We expect it to exhibit the same level of coherence as the fixed-order matrix elements, at least at leading colour (LC), since its QCD antenna functions and corresponding phase-space factorisations explicitly incorporate the soft-eikonal function for all possible (LC) colour flows. Of particular relevance to this study is its coherent treatment of "initial-final" (IF) colour flows.

- PYTHIA's default "simple shower" algorithm [11, 12], which implements $p_\perp$-ordered DGLAP evolution with dipole-style kinematics. For IF colour flows, however, the kinematic dipoles are not identical to the colour dipoles, and this can impact coherence-sensitive observables [51].

- PYTHIA's "simple shower" with the dipole-recoil option [16]. Despite its name, this not only changes the recoil scheme; in fact, it replaces the two independent DGLAP evolutions of IF dipoles by a coherent, antenna-like, dipole evolution, while keeping the DGLAP evolution of other dipoles unchanged. This option should therefore lead to radiation patterns exhibiting a similar level of coherence as VINCIA.

Ordinarily, PYTHIA would of course also add decays of the Higgs boson, and any final-state radiation associated with that. However, as a colour-singlet scalar with $\Gamma_H \ll \Lambda_{QCD}$ and $\Gamma_H/M_H \sim \mathcal{O}(10^{-5})$, its decay can be treated as factorised from the production process to a truly excellent approximation. For the purpose of this study, we therefore keep the Higgs boson stable, to be able to focus solely on the radiation patterns of the VBF production process itself, without the complication of decay products in the central region.

For all of the shower algorithms, we retain PYTHIA's default PDF choice[3], regardless of which PDF set was used to generate the hard process. This is done to remain consistent with the default shower tunings [52] and due to the better-controlled backwards-evolution properties of the default set [53].

Per default, the shower starting scale is chosen to be the factorisation scale of the hard process,

$$\mu_{PS}^2 = \mu_F^2. \tag{7}$$

In VINCIA, this scale can be varied by a multiplicative "fudge" factor, controlled by Vincia:pTmaxFudge,

$$\mu_{PS}^2 = k_{fudge}\, \mu_F^2,$$

---

[3]NNPDF23_lo_as_0130_qed.

while in PYTHIA, the starting scales of the initial-state and final-state showers can be varied independently,

$$\mu^2_{\text{PS,FSR}} = k_{\text{fudge,FSR}} \mu^2_{\text{F}},$$
$$\mu^2_{\text{PS,ISR}} = k_{\text{fudge,ISR}} \mu^2_{\text{F}},$$

controlled by `TimeShower:pTmaxFudge` and `SpaceShower:pTmaxFudge`, respectively.

In a similar vein, the strong coupling in the shower is evaluated at the shower $p_{\text{T}}$-scale[4], modified by renormalisation-scale factors $k_{\text{ren}}$. In PYTHIA, the strong coupling at the $Z$ mass is set to $\alpha_{\text{S}}(M_Z) = 0.1365$ and independent scale factors for ISR and FSR are implemented,

$$\alpha_{\text{S}}^{\text{Pythia,FSR}}(p^2_{\perp\text{evol,FSR}}) = \alpha_{\text{S}}^{\overline{\text{MS}}}(k_{\text{R,FSR}} \, p^2_{\perp\text{evol,FSR}}),$$
$$\alpha_{\text{S}}^{\text{Pythia,ISR}}(p^2_{\perp\text{evol,ISR}}) = \alpha_{\text{S}}^{\overline{\text{MS}}}(k_{\text{R,ISR}} \, p^2_{\perp\text{evol,ISR}}).$$

These can be set via `TimeShower:renormMultFac` and `SpaceShower:renormMultFac`, respectively, and are unity by default. The transverse-momentum evolution variables $p^2_{\perp\text{evol,FSR}}$ and $p^2_{\perp\text{evol,ISR}}$ are defined as in [11].

For VINCIA, on the other hand, a more refined choice can be made with separate renormalisation factors being implemented for (initial- and final-state) emissions, (initial- and final-state) gluon splittings, and (initial-state) quark conversions. These have the default settings:

$$k_{\text{R,Emit}}^{\text{F}} = 0.66, \quad k_{\text{R,Split}}^{\text{F}} = 0.8,$$
$$k_{\text{R,Emit}}^{\text{I}} = 0.66, \quad k_{\text{R,Split}}^{\text{I}} = 0.5, \quad k_{\text{R,Conv}}^{\text{I}} = 0.5,$$

which can be set via the parameters

```
Vincia:renormMultFacEmitF
Vincia:renormMultFacSplitF
Vincia:renormMultFacEmitI
Vincia:renormMultFacSplitI
Vincia:renormMultFacConvI.
```

Additionally, VINCIA uses the CMW scheme [54] (while PYTHIA does not), i.e., it evaluates the strong coupling according to

$$\alpha_{\text{S}}^{\text{CMW}} = \alpha_{\text{S}}^{\overline{\text{MS}}} \left( 1 + \frac{\alpha_{\text{S}}^{\overline{\text{MS}}}}{2\pi} \left[ C_A \left( \frac{67}{18} - \frac{\pi^2}{6} \right) - \frac{5 n_f}{9} \right] \right), \tag{8}$$

where $\alpha_{\text{S}}^{\overline{\text{MS}}}(M_Z) = 0.118$, so that

$$\alpha_{\text{S}}^{\text{Vincia}}(p^2_\perp) = \alpha_{\text{S}}^{\text{CMW}}(k_{\text{R}} \, p^2_\perp), \tag{9}$$

with the VINCIA evolution variable as defined in [27].

## 2.3 Matching and Merging

In the following, we will briefly review the defining features of the POWHEG NLO matching and the CKKW-L merging schemes we will use in this study. In particular, we will focus on the technicalities and practicalities to ensure a consistent use. Detailed reviews of the POWHEG schemes can for instance be found in [55] and [56]. The CKKW-L scheme is explained in detail in [39] and its extension to the VINCIA sector shower in [40].

---

[4]We refer to the argument of the strong coupling used in the shower as the shower renormalisation scale.

### 2.3.1 POWHEG Matching

In the POWHEG formalism, events are generated according to the inclusive NLO cross section with the first emission generated according to a matrix-element corrected no-emission probability.

Since the shower kernels in the POWHEG no-emission probability are replaced by the ratio of the real-radiation matrix element to the Born-level one, it is independent of the shower it will later be matched to. It is, however, important to stress that generally, the POWHEG ordering variable will not coincide with the ordering variable of the shower. Starting a shower with a different ordering variable at the POWHEG scale of the first emission might thus lead to over- or undercounting emissions. A simple method to circumvent this was presented in [57]. There, the shower is started at the phase space maximum (a so-called "power shower" [58]) and emissions harder than the POWHEG one are vetoed until the shower reaches a scale below the scale of the first emission. For general ordering variables, there is, however, no guarantee that once the shower falls below the scale of the POWHEG emission it will not generate a harder emission later on in the evolution. This is especially important if the shower is not ordered in a measure of hardness but e.g. in emission angles, such as the HERWIG $\tilde{q}$ shower [59]. In these cases, it is advisable to recluster the POWHEG emission and start a truncated and vetoed shower off the Born state [48], see also [60–62] for the use of truncated showers in merging schemes. This scheme also avoids the issue that in vetoed showers, all emissions in the shower off a Born+1-jet state are compared against the POWHEG emission as if they were the first emission themselves. But from the point of view of kinematics and colour they will still be the second, third, etc.

However, since all showers we consider here are ordered in a notion of transverse momentum, it shall suffice for our purposes to use the simpler "vetoed power shower" scheme. To this end, we have amended the existing POWHEG user hook for PYTHIA's showers by a dedicated one for POWHEG+VINCIA, which has been included in the standard release of PYTHIA starting from version 8.306; see appendix A for detailed instructions.

For both PYTHIA and VINCIA, we use a vetoed shower with the POWHEG $p_T$ and $d_{ij}$ definitions, corresponding to the mode `POWHEG:pTdef = 1`. We define the POWHEG scale with respect to the radiating leg and use PYTHIA's definition of emitter and recoiler, corresponding to the modes `POWHEG:pTemt = 0` and `POWHEG:emitted = 0`. Per default, we choose to define the scale of the POWHEG emission by the minimum $p_T$ among all final-state particles, i.e. use `POWHEG:pThard = 2`, according to the suggestion in [63]. As an estimate of the uncertainty of this choice, we vary the $p_{T,hard}$ scale to be the LHEF scale and the $p_T$ of the POWHEG emission, corresponding to the modes `POWHEG:pThard = 0` and `POWHEG:pThard = 1`, respectively.

The purpose of these settings is to ensure maximally consistent scale definitions while not reverting to the (more involved) "truncated and vetoed shower" scheme mentioned above. While we deem the choices made here appropriate for the case at hand they remain ambiguous, effectively introducing systematic matching uncertainties into the (precision) calculation. As a means of estimating these uncertainties, we will discuss the influence of the $p_{T,hard}$ scale setting on physical observables below in section 3.

### 2.3.2 CKKW-L Merging

Multi-leg merging schemes aim at correcting parton shower predictions away from the soft and collinear regions. In the CKKW-L merging scheme [39], multiple inclusive tree-level event samples are combined to a single inclusive one by introducing a (somewhat arbitrary) "merging scale" $t_{MS}$ which separates the matrix-element ($t > t_{MS}$) from the parton-shower ($t < t_{MS}$) region. In this way, over-counting of emissions is avoided while accurate parton-shower resummation in logarithmically enhanced regions and leading-order accuracy in the regions of

hard, well-separated jets is ensured if the merging scale is chosen appropriately.

The missing Sudakov suppression in higher-multiplicity configurations is calculated post-facto by the use of truncated trial showers between the nodes of the most probable "shower history". In this context, the shower history represents the sequence of intermediate states the parton shower at hand would (most probably) have generated to arrive at the given $n$-jet state. Usually, this sequence is constructed by first finding all possible shower histories and subsequently choosing the one that maximises the branching probability, i.e., the product of branching kernels and the Born matrix element. As we employ this scheme with VINCIA's sector shower, a few comments are in order. The objective of the sector shower is to replace the probabilistic shower history by a deterministic history, governed by the singularity structure of the matrix element. This means that at each point in phase space only the most singular branching contributes. In the shower, this is ensured by vetoing any branchings that do not abide by this; in the merging, this results in a faster and less resource-intensive algorithm, as it is no longer required to generate a large number of possible histories. Details and subtleties of VINCIA's sectorised CKKW-L implementation can be found in [40].

The CKKW-L merging scheme is in principle implemented for all showers in PYTHIA 8.3. However, the intricate event topology of VBF processes currently prohibits the use of PYTHIA's default merging implementation[5]. We hence limit ourselves to study the effect of merging with VINCIA, and have adapted VINCIA's CKKW-L implementation [40] so that VBF processes are consistently treated. Specifically, the flag `Vincia:MergeVBF = on` should be used, which restricts the merging to only consider shower histories that retain exactly two initial-final quark lines. As a consequence, there must not be any "incomplete histories" (histories that do not cluster back to a VBF Born configuration); this should be guaranteed as long as the input event samples are of the VBF type only and no QED or EW emissions are generated. A complete list of relevant settings for the use of VINCIA's CKKW-L merging is collected in appendix B.

## 2.4 Analysis

We use the anti-$k_\mathrm{T}$ algorithm [64] with $R = 0.4$, as implemented in the FASTJET [65] package, to cluster jets in the range,

$$p_\mathrm{T} > 25 \text{ GeV}, \quad |\eta| < 4.5.$$

In addition, we employ typical VBF cuts to ensure that the two "tagging jets" are sufficiently hard, have a large separation in pseudorapidity, and are located in opposite hemispheres:

$$m_{j_1,j_2} \geq 600 \text{ GeV}, \quad |\Delta\eta_{j_1,j_2}| \geq 4.5, \quad \eta_{j_1} \cdot \eta_{j_2} \leq 0.$$

We consider the following observables:

- **Pseudorapidity Distributions:** at the Born level, the two tagging jets already have non-trivial pseudorapidity distributions. These are sensitive to showering chiefly via recoil effects and via the enhancement of radiation towards the beam directions. The third (and subsequent) jets are of course directly sensitive to the generated emission spectra. To minimise contamination from final-state radiation off the tagging jets, we also consider the pseudorapidity of the radiated jet(s) relative to the midpoint of the two tagging jets,

$$\eta_{j_i}^* = \eta_{j_i} - \eta_0, \tag{10}$$

  with the midpoint defined by:

$$\eta_0 = \tfrac{1}{2}(\eta_{j_1} + \eta_{j_2}). \tag{11}$$

---

[5]We note that a technical fix for this was available in PYTHIA 8.242 and will become available again in PYTHIA 8.3 in the future.

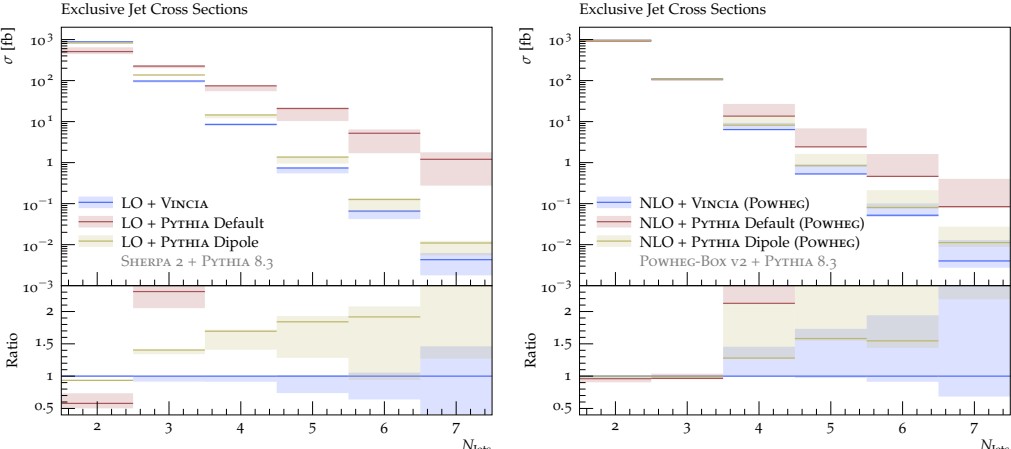

Figure 2: Exclusive jet cross sections at LO+PS (*left*) and Powheg NLO+PS (*right*) accuracy. The bands are obtained by a variation of the default shower starting scale by a factor of two or the variation of the hard scale, respectively.

- **Transverse Momentum Distributions:** we expect coherence effects for the radiated jets ($i > 2$) to be particularly pronounced for radiation that is relatively soft in comparison to the characteristic scale of the hard process. Conversely, the transverse momenta of the two tagging jets should mainly be affected indirectly, via momentum-conservation (recoil) effects.

- **Scalar Transverse Momentum Sum:** as a more inclusive measure of the summed jet activity in the central rapidity region, we consider the scalar transverse momentum sum of all reconstructed jets (defined as above, i.e., with $p_T > 25$ G*e*V),

$$H_T = \sum_j |p_{T,j}|, \tag{12}$$

in two particular regions:

  – in the central rapidity region, $\eta \in \left[-\frac{1}{2}, +\frac{1}{2}\right]$
  – around the midpoint of the tagging jets, $\eta^* \in \left[-\frac{1}{2}, +\frac{1}{2}\right]$, cf eq. (10).

We point out that, due to the way it is constructed, the second of these regions is not sensitive to the tagging jets, as it is not possible for them to fall within this region. Unlike the previous two observables, $H_T$ is sensitive to the overall radiation effect in the given region, not just that of a certain jet multiplicity. As such, we expect $H_T$ to give a measure of the all-orders radiation effects.

The analysis is performed using the Rivet analysis framework [66,67] and based on the one used in [8].

# 3 Results

In this section, we present the main results of our study based on the setup described in the last section. In fig. 2, the exclusive jet cross sections for up to 7 jets are shown at LO+PS and NLO+PS (via the Powheg scheme) accuracy at the Born level. While there are very large differences between the three shower predictions at the leading order, there is good agreement between the NLO+PS predictions at least for the 2- and 3-jet cross sections.

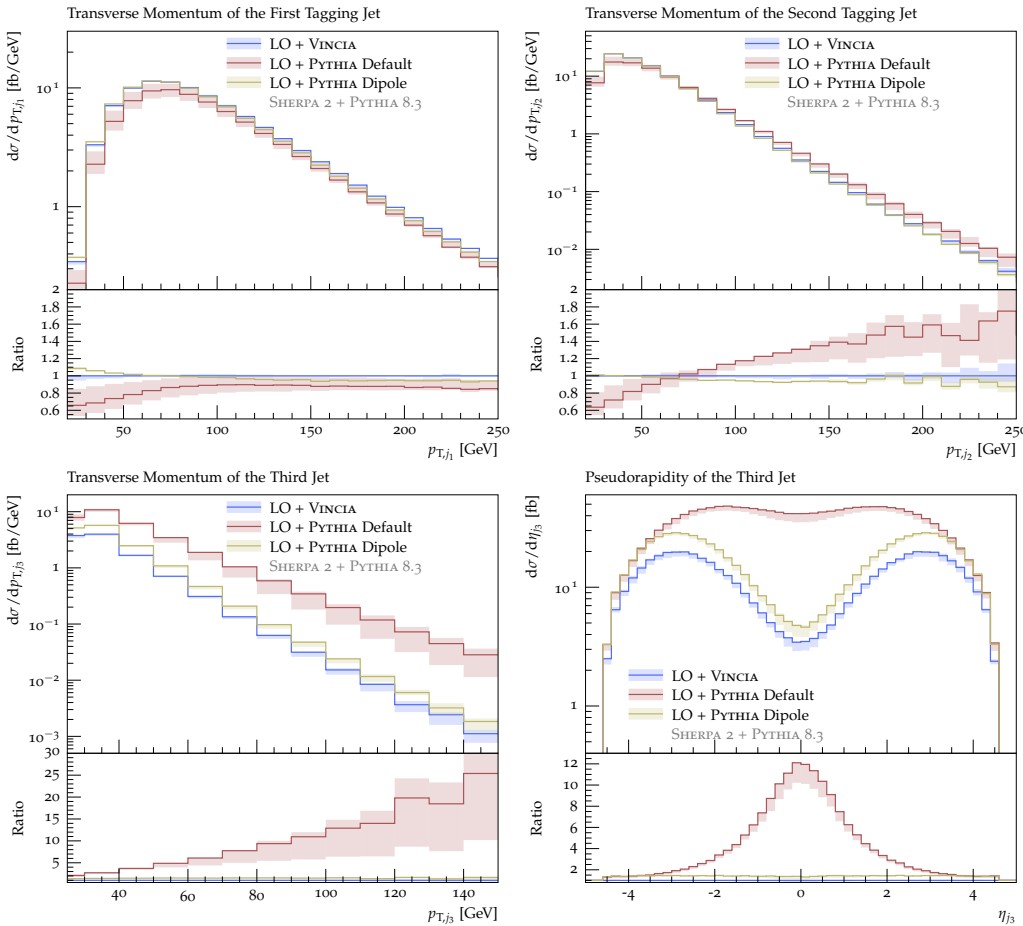

Figure 3: Transverse momentum of the first tagging jet (*top left*), second tagging jet (*top right*), third jet (*bottom left*), and pseudorapidity of the third jet (*bottom right*) at LO+PS accuracy. The bands are obtained by a variation of the default shower starting scale by a factor of two.

## 3.1 Leading Order

It is instructive to start by studying the properties of the baseline leading-order + shower calculations, without including higher fixed-order corrections.

We use the leading-order event samples generated with SHERPA and by default let the factorisation scale $\mu_F^2$ define the shower starting scale. As a way to estimate the uncertainty associated with this choice, we vary the shower starting scale $\mu_{PS}^2$ by a factor $k_{fudge} \in \left[\frac{1}{2}, 2\right]$, $\mu_{PS}^2 = k_{fudge}\mu_F^2$. Strictly speaking, shower starting scales not equal to the factorisation scale lead to additional PDF ratios in the no-branching probabilities generated by the shower, but for factor-2 variations these are consistent with unity (since the PDF evolution is logarithmic) and we therefore neglect them. Compared to the shower starting scale, variations of the shower renormalisation scale only have a marginal effect and are therefore not shown here. As we are primarily concerned with the shower radiation patterns, we do not vary the scales in the fixed-order calculation. The effect of those variations have been studied extensively in the literature before, cf. e.g. [8, 18].

In fig. 3, the transverse momentum distributions of the two tagging jets and as well as the transverse momentum and pseudorapidity distributions of the third-hardest jet are shown. While the tagging jet $p_T$ spectra agree well between VINCIA and PYTHIA with dipole recoil, differences are visible for the third-jet observables, with similar shapes but a slightly larger

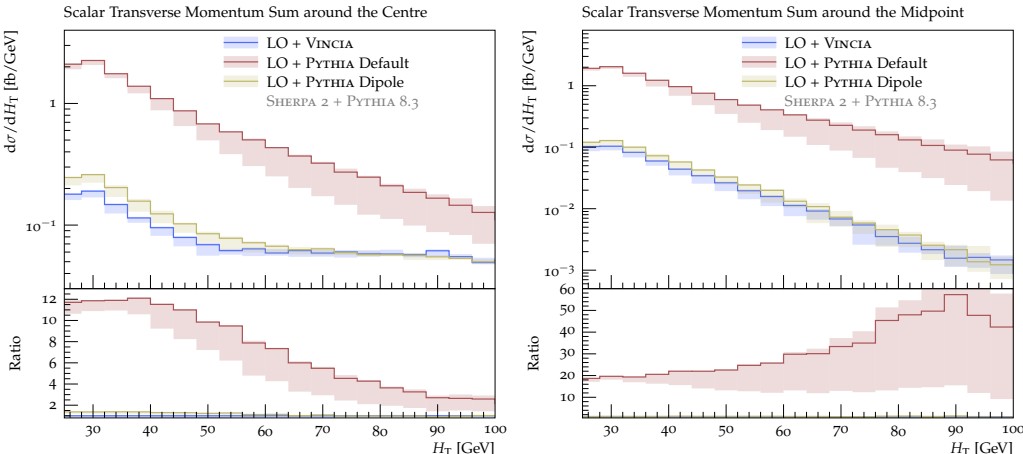

Figure 4: Scalar transverse momentum sum in the central rapidity region (*left*) and around the rapidity midpoint of the tagging jets (*right*) at LO+PS accuracy. The bands are obtained by a variation of the default shower starting scale by a factor of two.

rate produced by the PYTHIA dipole-recoil shower. The distributions obtained with the PYTHIA default shower, on the other hand, neither agree in shape nor in the rate with the other two. In fact, almost no suppression of radiation in the central-rapidity region is visible and the shower radiation appears at a much higher transverse momentum scale. The high emission rate in the default shower also implies that the tagging jets receive much larger corrections with this shower than with the others, as evident from the tagging-jet $p_T$ distributions.

Figure 4 shows the $H_T$ distributions in the previously defined central and midpoint regions. As for the third-jet pseudorapidity and transverse-momentum distributions, there is only a minor disagreement between PYTHIA dipole-recoil shower and VINCIA, while PYTHIA's DGLAP shower generates significantly more radiation in both regions.

For all observables considered here, we also note that the variation of the shower starting scale has a much more pronounced effect on the PYTHIA default shower than on VINCIA or on PYTHIA when the dipole-recoil option is enabled. Moreover, the starting-scale variation affects the $p_T$ distribution of the third jet more than it does the pseudorapidity distribution. This indicates that, while a tailored shower starting scale for the default shower might be able to mimic the phase space-suppression of the dipole/antenna showers to some extent, this would not by itself be sufficient to represent the dipole-antenna emission pattern of the third jet.

We close this subsection by comparing showers off our externally generated Born-level VBF events (i.e., ones generated by SHERPA and passed to PYTHIA for showering) to showers off internally generated ones (i.e., ones generated by PYTHIA's `HiggsSM:ff2Hff(t:WW)` and `HiggsSM:ff2Hff(t:ZZ)` processes). This is intended as a cross check for any effects caused by differences in how PYTHIA treats external vs internal events. For instance, for external events, the external generator is responsible not only for computing the hard cross section but also for setting the shower starting scale, via the HDF5 `scales` dataset (equivalent to the Les Houches SCALUP parameter [68, 69]). For our VBF events, the choice made in SHERPA is identical to the factorisation scale eq. (3),

$$\text{SHERPA VBF events:} \qquad \mu_{\text{PS}}^2 \equiv \mu_{\text{F}}^2 = \frac{\hat{H}_{\text{T}}^2}{4} = \frac{1}{4} \left( \sum_j p_{\text{T},j} + \sqrt{M_{\text{H}}^2 + p_{\text{T},\text{H}}^2} \right)^2 .$$

For internally generated VBF events, PYTHIA's choice of the factorisation scale, and thereby also the shower starting scale, is designed to reflect the off-shellness of the two virtual-boson

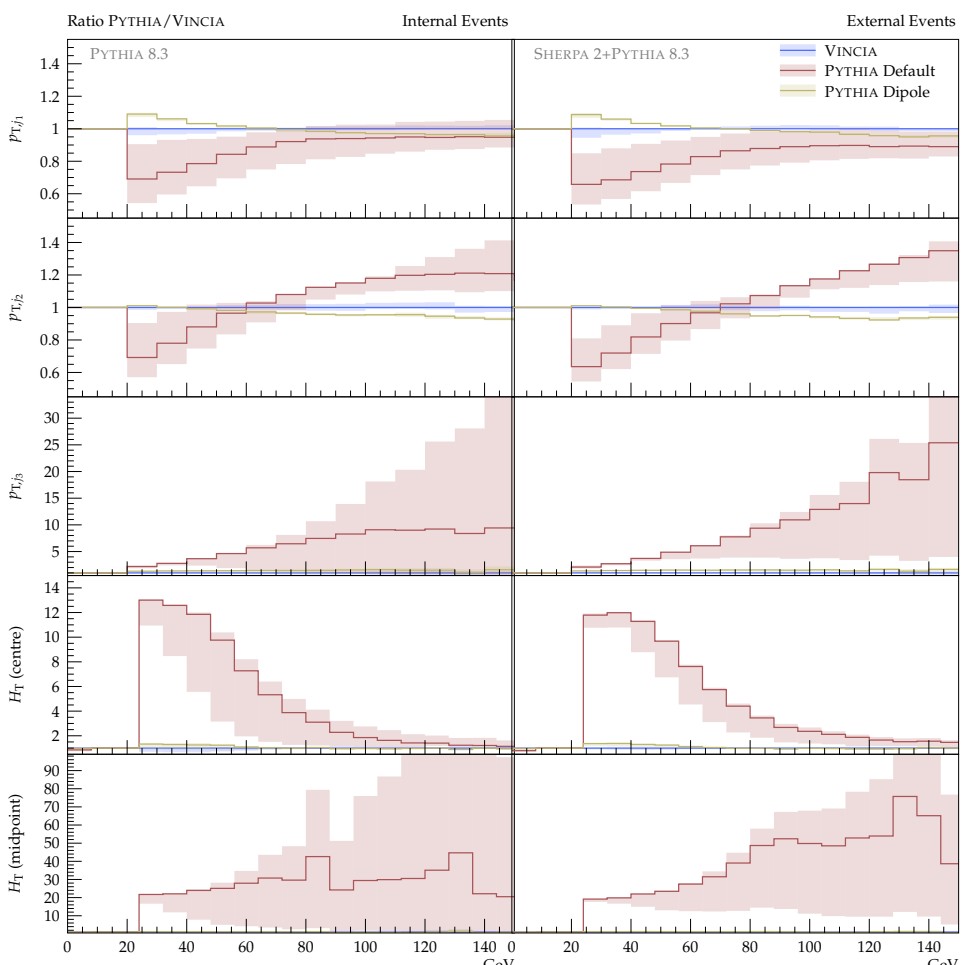

Figure 5: Ratio of PYTHIA to VINCIA at LO+PS accuracy, comparing internal (*left*) and external (*right*) events. The bands are obtained by a variation of the factorisation scale (internal events) and shower starting scale (external events) by a factor of two.

$t$-channel propagators, cf. eq. (5),

$$\text{PYTHIA VBF events:} \qquad \mu_{\text{PS}}^2 \equiv \mu_{\text{F}}^2 = \sqrt{m_{\text{T},V_1}^2 m_{\text{T},V_2}^2} \equiv \sqrt{(M_{V_1}^2 + p_{\text{T},q_1}^2)(M_{V_2}^2 + p_{\text{T},q_2}^2)}.$$

This choice ensures that the factorisation scale and shower starting scale will always be at least of order $M_V^2$ even when the outgoing quarks have low $p_T \ll M_V$, while for very large $p_T$ values, it asymptotes to the geometric mean of the quark $p_T$ values. While the minimum of the SHERPA choice is of the same order, $\mathcal{O}(M_H) \sim \mathcal{O}(M_V)$, the large-transverse-momentum limit is considerably larger. The expectation is therefore that, in the absence of matching or merging corrections, SHERPA-generated Born events will lead to higher amounts of hard shower radiation than PYTHIA-generated ones.

In fig. 5, the ratio of the two PYTHIA showers to VINCIA is shown for the $p_T$ and $H_T$ spectra using (left) PYTHIA LO and (right) SHERPA LO events. We immediately note that, in the low-$p_\perp$ limit, the excess of soft radiation generated by PYTHIA's default shower (red) persists in both samples. In the high-$p_\perp$ regions, the agreement between the simple shower and the two dipole/antenna options (blue and yellow) tends to be best for PYTHIA's internal hard process. This likely originates from the lower value for the default shower starting scale in PYTHIA, which, as discussed above, imitates the propagator structure of the Born process as closely as possible and hence *should* to some extent set a natural boundary for strongly-ordered prop-

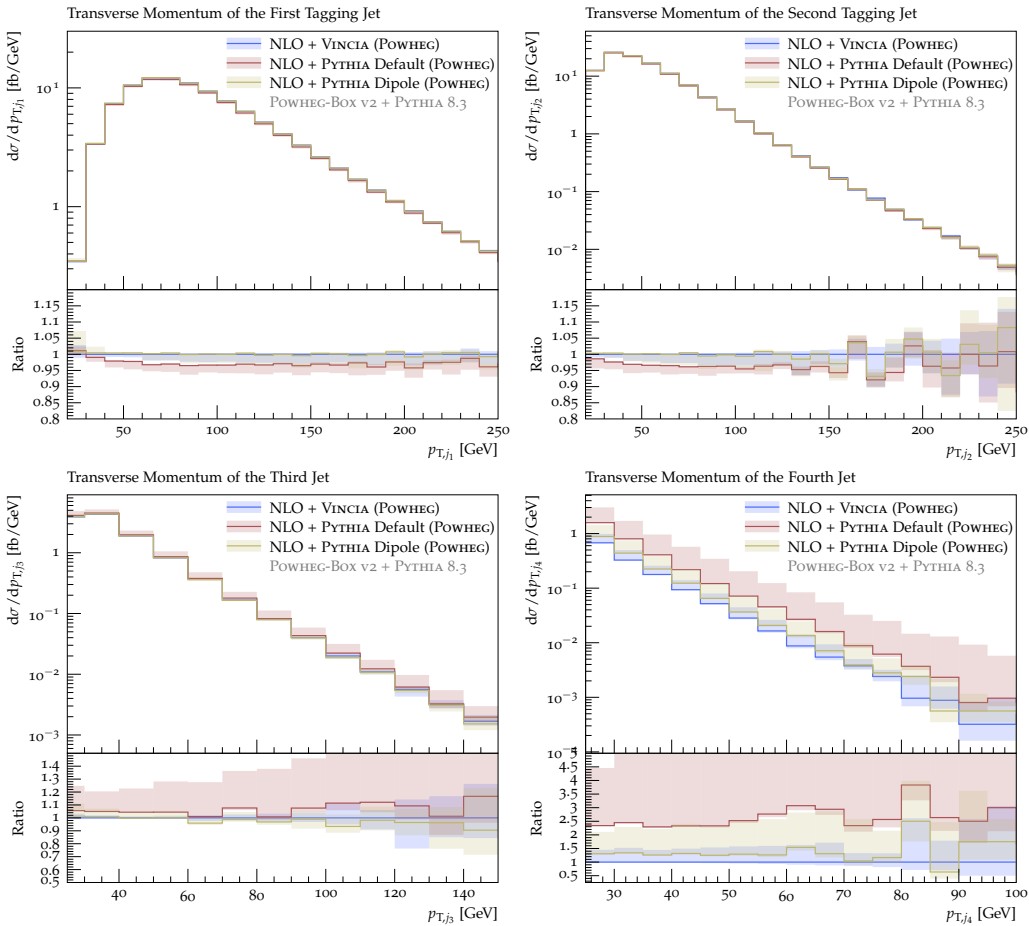

Figure 6: Transverse momentum of the first tagging jet (*top left*), second tagging jet (*top right*), third jet (*top left*), and fourth jet (*top right*) at NLO+PS accuracy in the POWHEG scheme. The bands are obtained by a variation of the hard scale in the vetoed showers as explained in the text.

agators in the shower. For the dipole/antenna showers, the sensitivity to the starting scale is far milder, as the relevant kinematic information is encoded in the dipole invariant masses independently of the choice of starting scale.

## 3.2 Next-to-Leading Order Matched

In fig. 6, the POWHEG-matched transverse momentum distributions of the four hardest jets are collected. In comparison to the LO+PS case discussed in the last section, it is directly evident that the Born-jet $p_T$ distributions are in good agreement between all three shower algorithms, including the default PYTHIA one, for which the tagging jet $p_T$ distributions undershoot the VINCIA curve only by an approximately constant factor of order of five per cent. After POWHEG matching, almost perfect agreement is found for the tagging-jet transverse momentum distributions obtained with VINCIA and PYTHIA with dipole recoil, as can be seen in fig. 8. The NLO corrections are, however, slightly smaller for the former. The scale choice of the POWHEG emission has only mild effects on all three showers for these tagging-jet observables.

Good agreement is also found between all three shower algorithms for the $p_T$ of the third jet, as shown in the bottom left panel of fig. 6. It must be noted that, again in the case of the PYTHIA default shower, this agreement is subject to appropriately vetoing harder emissions than the POWHEG one, which requires the definition of the POWHEG scale according to the minimal

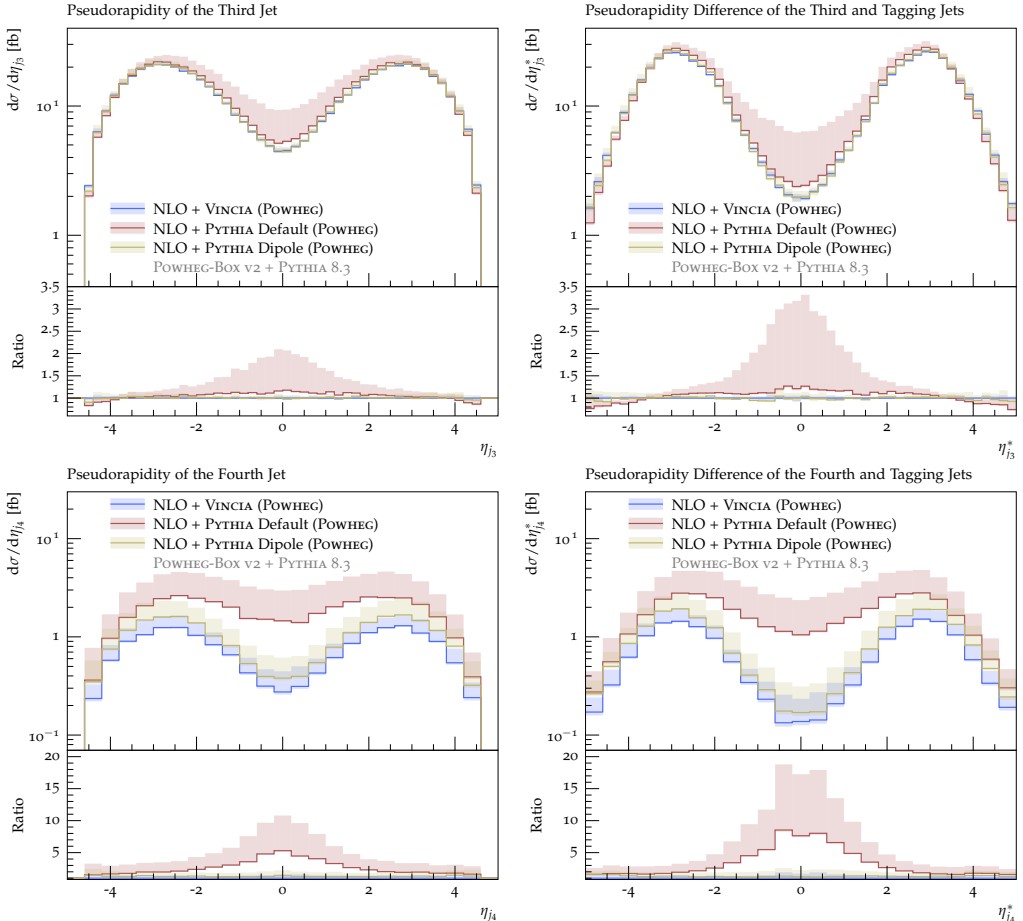

Figure 7: Pseudorapidity (*left column*) and relative rapidity to the tagging jets (*right column*) of the third jet (*top row*) and fourth jet (*bottom row*) at NLO+PS accuracy in the POWHEG scheme. The bands are obtained by a variation of the hard scale in the vetoed showers as explained in the text.

$p_T$ in the event, corresponding to the `POWHEG:pThard = 2` setting, cf. section 2.3.1. Other choices again lead to too hard third jets and heavily increased radiation in the central rapidity region, as can be inferred from the (relative) pseudorapidity distributions of the third jet in the top row of fig. 7, where the importance of a judicious POWHEG scale choice is especially visible. As for the tagging jet spectra, the agreement in both the third-jet transverse momentum and pseudorapidity predictions between VINCIA and the dipole-improved PYTHIA shower is almost perfect, as shown in fig. 9. While the correction (which in this case is essentially a LO matrix-element correction) is positive for VINCIA, it is negative for the dipole-improved PYTHIA shower. Moreover, in the case of VINCIA, this correction affects mostly the high-$p_T$ and the central-rapidity region, whereas for PYTHIA's dipole-improved shower, the correction is negligible at zero rapidity but bigger (and almost) constant at larger rapidities as well as for the transverse momentum.

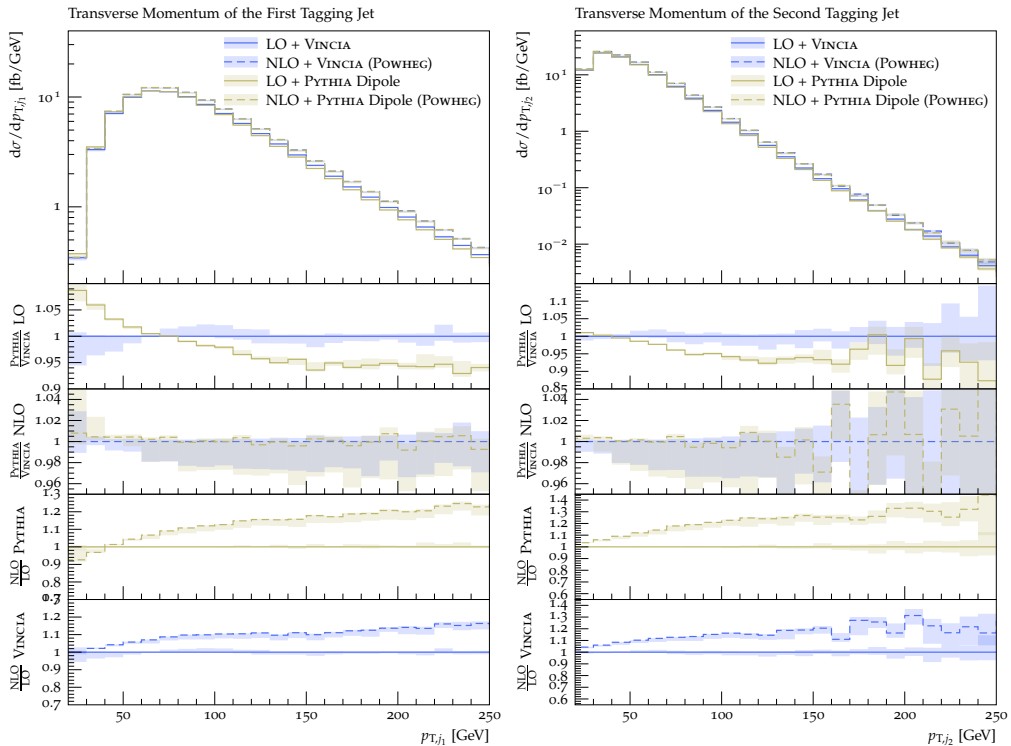

Figure 8: Detailed comparison of the PYTHIA dipole and VINCIA LO+PS and POWHEG NLO+PS predictions for the transverse momentum of the first tagging jet (*left*) and the second tagging jet (*right*).

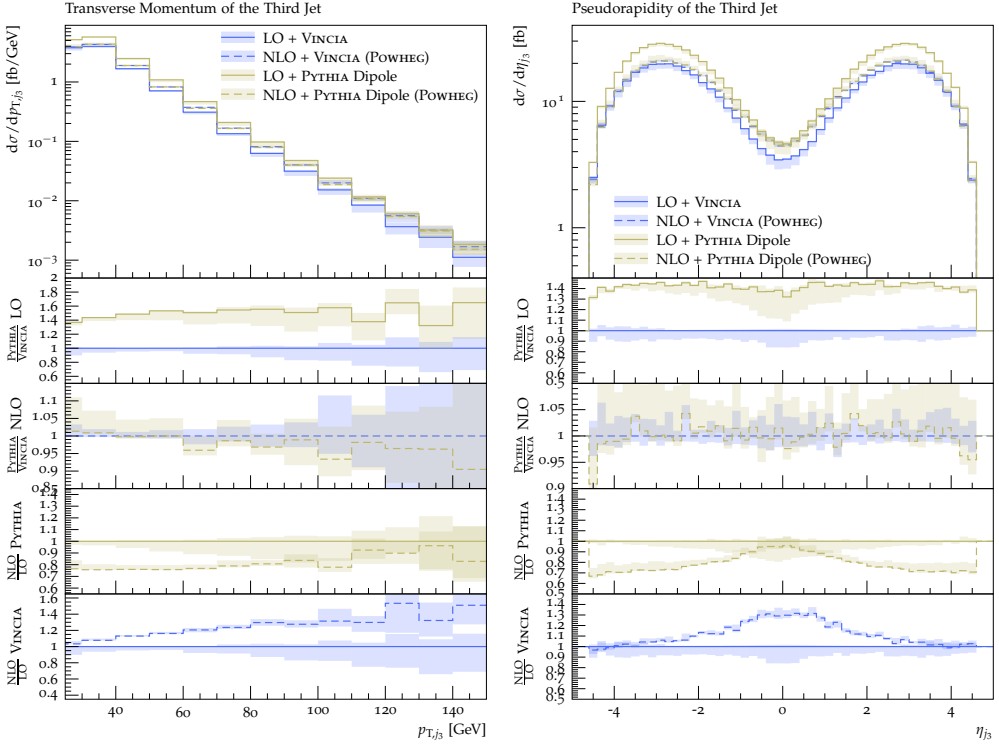

Figure 9: Detailed comparison of the PYTHIA dipole and VINCIA LO+PS and POWHEG NLO+PS predictions for the transverse momentum (*left*) and rapidity of the third jet (*right*).

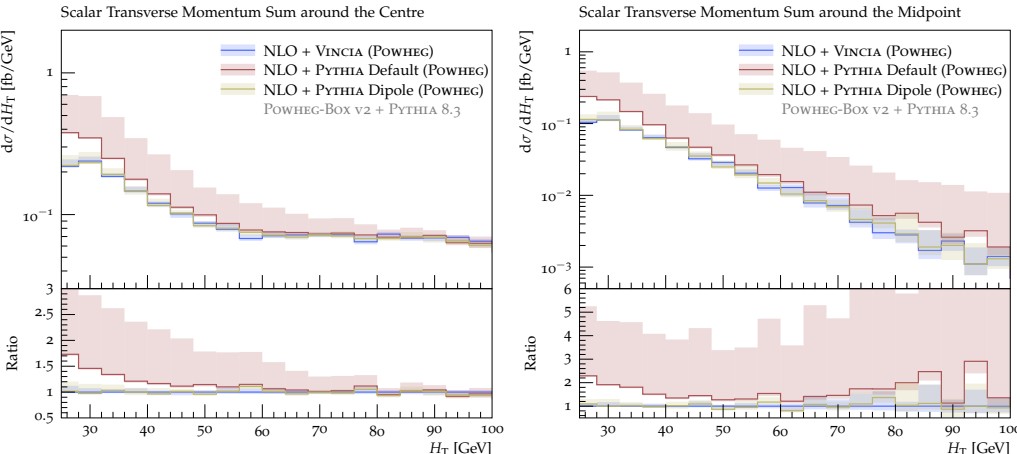

Figure 10: Scalar transverse momentum sum for $|\eta| < 0.5$ (*left*) and around the rapidity midpoint of the tagging jets (*right*) at NLO+PS accuracy in the POWHEG scheme. The bands are obtained by a variation of the hard scale in the vetoed showers as explained in the text.

The bottom right pane in fig. 6 and the bottom row in fig. 7 compare the $p_T$ and (relative) rapidity predictions of the three shower algorithms. While again rather good agreement in these distributions is found for the VINCIA shower and the dipole-improved PYTHIA shower, PYTHIA's default shower produces a harder spectrum, located more in the central pseudorapidity region. Here, it is worthwhile noting that for two-jet POWHEG matching, the emission of the fourth jet is uncorrected in either of the shower algorithms, so that the effects visible in these distributions are solely produced by the showers.

Lastly, fig. 10 shows the scalar transverse momentum for $|\eta| < 0.5$ (left) and around the tagging jet midpoint (right) in the POWHEG NLO+PS scheme. In both distributions, the three shower algorithms produce similar results for $H_T > 40$ G$e$V, while in the complementary region again only VINCIA and the dipole-improved PYTHIA shower agree. In this soft region, the default PYTHIA shower again predicts more radiation than the other two. As before, a variation of the POWHEG scale choice leads to significant effects in the predictions of PYTHIA's default shower, but has only mild effects on the dipole-improved shower and VINCIA.

### 3.3 Comparison of Matching and Merging

In figs. 11 to 13, we compare the VINCIA NLO-matched predictions presented in the last section to an $\mathcal{O}(\alpha_S)$ tree-level merged calculation using the CKKW-L scheme implemented for VINCIA. For the latter, we include the exclusive zero-jet and inclusive Sudakov-weighted 1-jet predictions in the plots (dashed lines).

The uncertainty bands of the merged predictions (labelled VINCIA MESS $\mathcal{O}(\alpha_S)$) are obtained by a variation of the shower renormalisation scale as per section 2.2. As VINCIA's merging implementation reweights event samples by a ratio of the strong coupling as used in the shower to the one used in the fixed-order calculation, this variation effectively amounts to an intertwined scale variation of the hard process as well. The uncertainty bands of the NLO-matched calculation are obtained by the variation of the $p_{\perp,\text{hard}}$ scale as in the previous section.

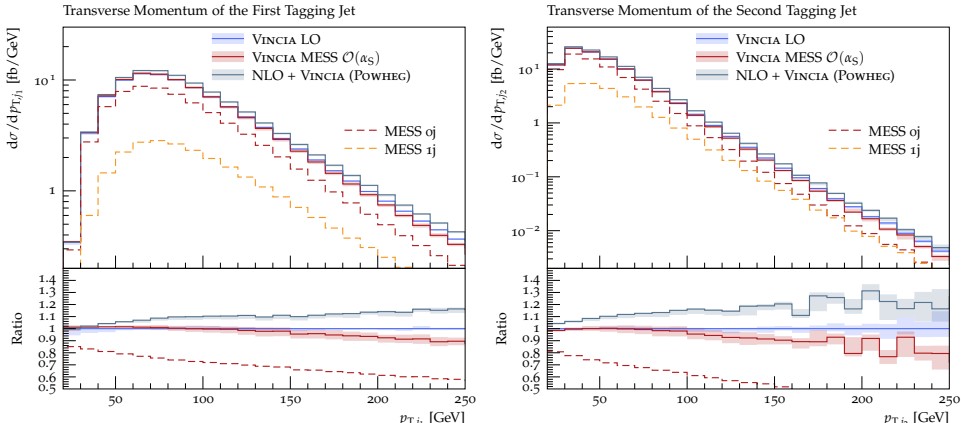

Figure 11: Comparison between LO+PS, POWHEG NLO+PS, and CKKW-L-merged predictions for the transverse momentum of the first (*left*) and second (*right*) tagging jet.

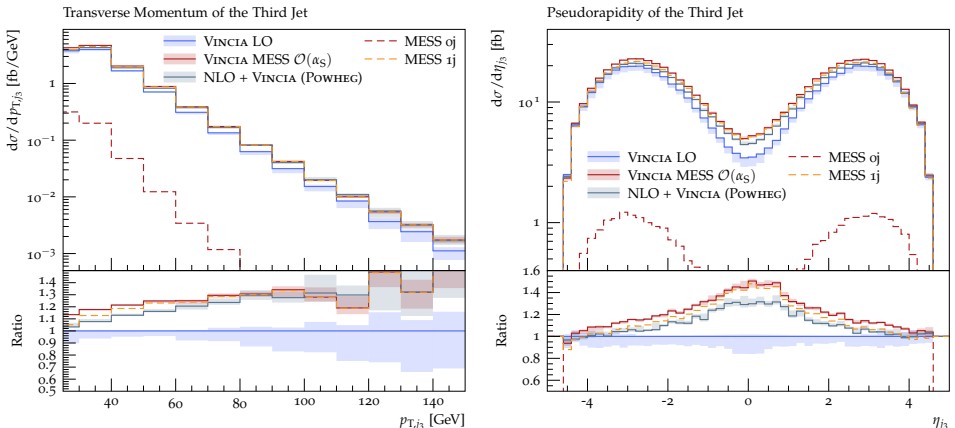

Figure 12: Comparison between LO+PS, POWHEG NLO+PS, and CKKW-L-merged predictions for the transverse momentum (*left*) and pseudorapidity (*right*) of the third jet.

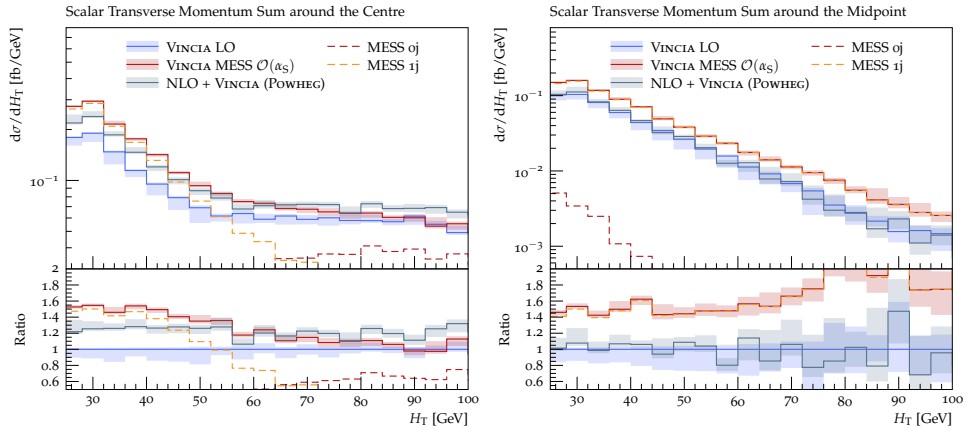

Figure 13: Comparison between LO+PS, POWHEG NLO+PS, and CKKW-L-merged predictions for the scalar transverse momentum sum for $|\eta| < 0.5$ (*left*) and around the pseudorapidity midpoint of the tagging jets (*right*).

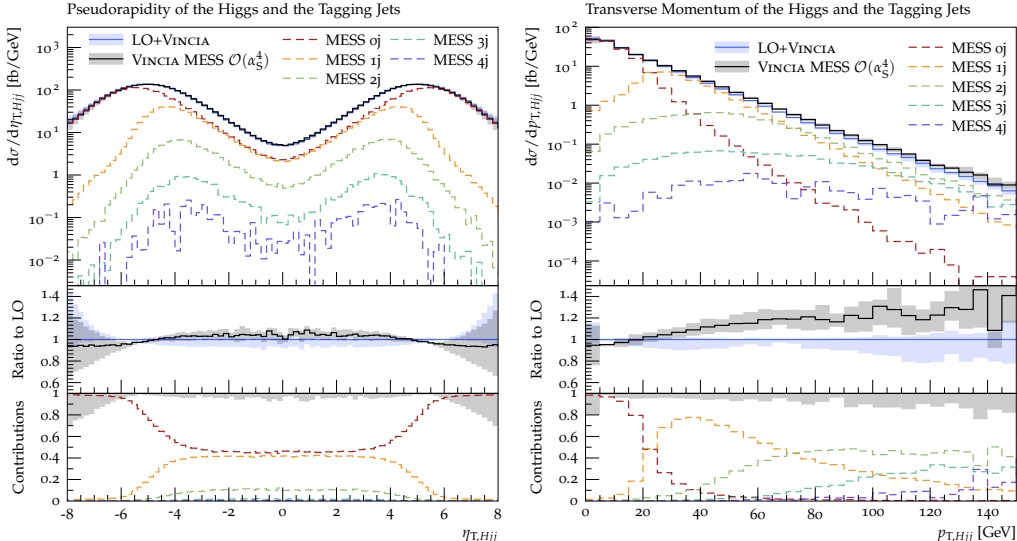

Figure 14: Tree-level merged predictions with up to four additional jets for the pseudorapidity (*left*) and transverse momentum (*right*) of the Higgs and tagging jets system.

Taking into account their respective accuracies, we observe good agreement between the matched and the merged predictions for the transverse momentum and pseudorapidity spectra. We expect the small differences that are visible to trace back mainly to the lack of unitarity in the CKKW-L scheme. This explanation is supported by the fact that the merged calculation overshoots the matched ones and that e.g. for the $p_{T,j_3}$ distribution, the inclusive Sudakov-reweighted 1-jet contribution already agrees in shape and magnitude with the matched distributions, while the exclusive zero-jet contributions only adds to the rate, i.e overall normalisation. In addition, we wish to note again that the mismatch of the POWHEG and VINCIA ordering variables is only treated approximately via the use of vetoed showers, while the correct shower history is taken into account in the merged calculation. Furthermore, we have used two different renormalisation and factorisation scales in the two calculations. Because the renormalisation scale variation in VINCIA's merging affects the renormalisation scale of the hard process, as alluded to above, the renormalisation scale mismatch is covered to some degree by the scale variations in the merging.

The situation is different for the $H_T$ distributions, cf. fig. 13. In the merged calculation, more soft radiation is predicted in the central pseudorapidity region than in the matched one. The distribution is solely governed by the one-jet sample there, while the zero-jet sample contributes significantly above 60 GeV only. In the midpoint region, however, the merged calculation predicts the same shape as the matched one, but with an overall bigger rate. Barely any contribution stems from the exclusive zero-jet sample in this observable. This confirms the properties of the two $H_T$ observables mentioned in section 2.4. When the observable is defined over the central rapidity region, it is sensitive to the radiation of the third jet in the soft region, i.e. for $H_T \lesssim 60$ GeV, but becomes sensitive to the tagging jets in the complementary hard region, i.e., above around 60 GeV. In contrast, defining the observable over the region around the pseudorapidity midpoint of the two tagging jets cleans it from almost all contributions stemming from the Born configuration (only a tiny contribution from soft radiation off the Born survives). Due to this property, the latter of the two definitions is particularly suited in the study of the radiation pattern regarding its coherence.

The comparison of NLO matching and $\mathcal{O}(\alpha_S)$ tree-level merging provides a strong cross check of both methods.

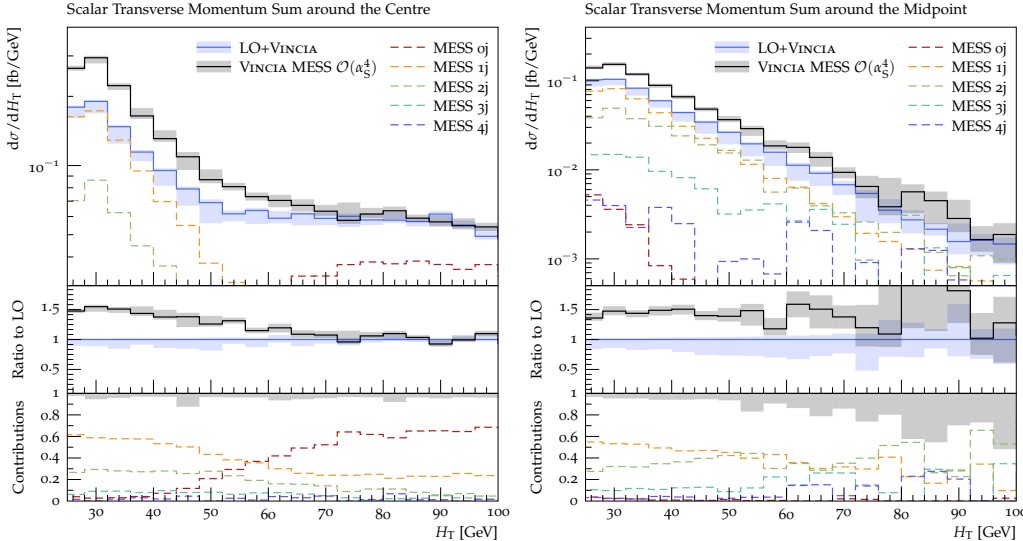

Figure 15: Tree-level merged predictions with up to four additional jets for the scalar transverse momentum sum in the central (*left*) and midpoint (*right*) pseudorapidity region.

## 3.4   Merged with up to Four Jets

In addition to the one-jet merged calculation of the last section, we here present a tree-level merged calculation with up to four additional jets (i.e., 6 jets in total when counting the tagging jets) using VINCIA's CKKW-L implementation. We consider the effect of additional hard jets on the spectra of the pseudorapidity and transverse momentum of the Higgs plus tagging jets system as well as the herein before mentioned scalar transverse momentum sum in the two pseudorapidity regions. The uncertainty bands of the merged calculation shown in the figures are obtained by a variation of the renormalisation scale prefactors $k_R$, c.f. section 2.2, in VINCIA's shower and merging, again effectively representing a variation of the renormalisation scale in the hard process as well, cf. section 3.3. As visible from fig. 15, the inclusion of additional hard jets does not change the pseudorapidity spectrum, but increases the rate of the transverse momentum spectrum in the high-$p_T$ region. This correction is exactly what is expected from a multi-jet merged calculation. The dashed lines in fig. 15 represent the different multi-jet contributions to the merged prediction. Again as expected, the Born sample dominates in the low-$p_T$ region and the one-jet sample in the region around 40 GeV, whereas higher multiplicities take over in the harder regions above $\sim$ 70 GeV. It is worth highlighting, however, that, at least in the region 70 GeV $\lesssim p_T \lesssim$ 150 GeV, the two-jet sample dominates with only sub-leading corrections from the three- and four-jet samples.

Figure 14 shows the $H_T$ distributions in the central and midpoint pseudorapidity regions defined in section 2.4. As for the one-jet merged prediction presented in section 3.3, the high-$H_T$ region is dominated by the Born sample, while for small $H_T$, the samples with additional jets define the shape. Although all samples with additional jets contribute to the central $H_T$ over the full shown spectrum, the three-jet sample (denoted 1$j$ in fig. 14) is the dominant extra-jet sample everywhere. Above approximately 60 GeV, the Born sample becomes the predominant one, highlighting again that this region is sensitive mainly to the tagging jets. Corrections from the multi-jet merging are negligible there.

As before, the situation is different in the midpoint region between the two tagging jets (right-hand pane in fig. 14). There, the Born sample has almost no impact ($< 5\%$) on the $H_T$ distribution and the one-jet sample (denoted 1$j$ in fig. 14) dominates in the region $\lesssim$ 70 GeV,

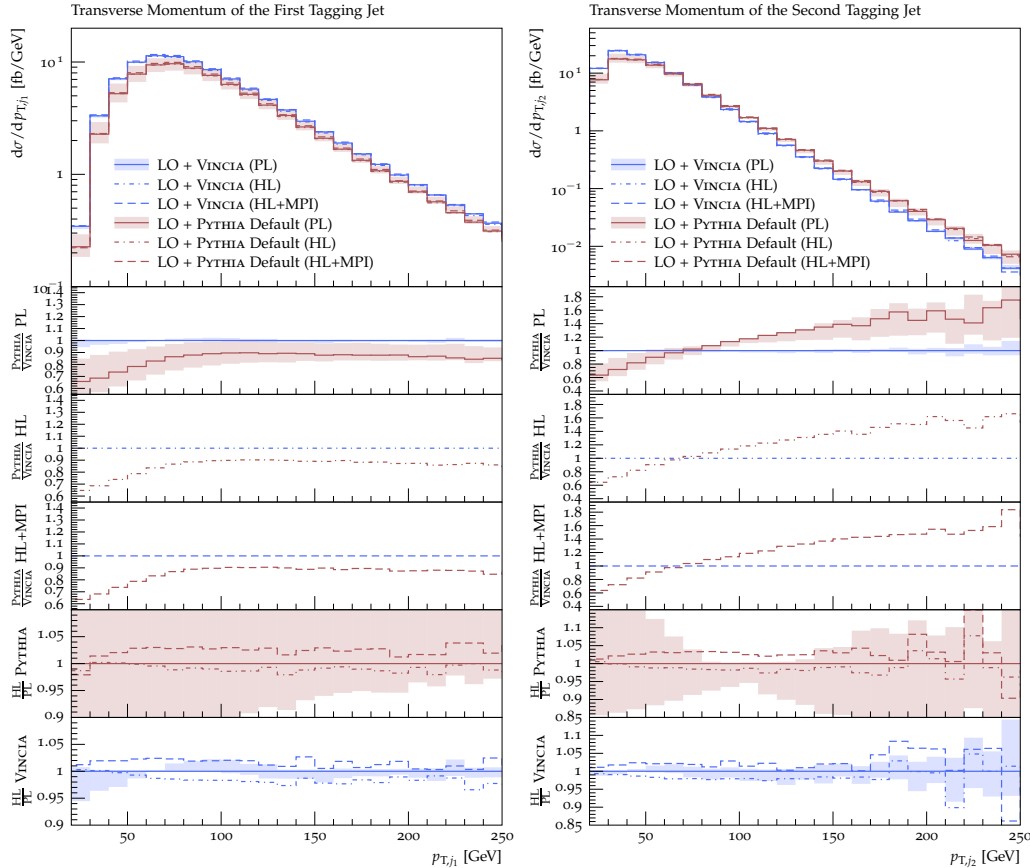

Figure 16: Detailed comparison of PYTHIA DGLAP and VINCIA LO+PS predictions at parton-level, hadron-level, and hadron-level plus MPI for the transverse momentum of the first tagging jet (*left*) and the second tagging jet (*right*).

while the two-jet sample (denoted $2j$ in fig. 14) does in the region 70 G$e$V $\lesssim H_T \lesssim$ 100 G$e$V. This emphasises the finding of the last section that the midpoint $H_T$ is clean of contributions from the tagging jets and therefore more relevant in the study of coherence effects in QCD radiation.

### 3.5 Hadronisation and Multi-Parton Interactions

Although we focused on the parton level throughout this study, we wish to close by estimating the size of non-perturbative corrections arising from hadronisation, fragmentation, and multi-parton interactions. To this end, we employ PYTHIA's string fragmentation and interleaved MPI model [11] using the default PYTHIA [52] and VINCIA [27] tunes.

Figures 16 to 18 compare PYTHIA's simple shower and VINCIA predictions on the parton level, hadron level, and hadron level with MPIs at LO+PS accuracy. As expected from the cuts employed in our analysis, cf. section 2.4, the inclusion of non-perturbative effects in either of the two simulations has only a negligible effect on most observables studied here, although the discrepancy between the two showers is slightly mitigated. A notable exception are the VINCIA predictions for the $H_T$ in the two pseudorapidity regions defined in section 2.4, for which the inclusion of MPIs leads to a substantial excess in radiation in the soft region. This means, that in those regions the coherent suppression of radiation by VINCIA is overwhelmed by the soft radiation off secondary (non-VBF-like) interactions, at least with our set of cuts. It should be noted here that firstly, this excess is not visible in the distributions obtained with

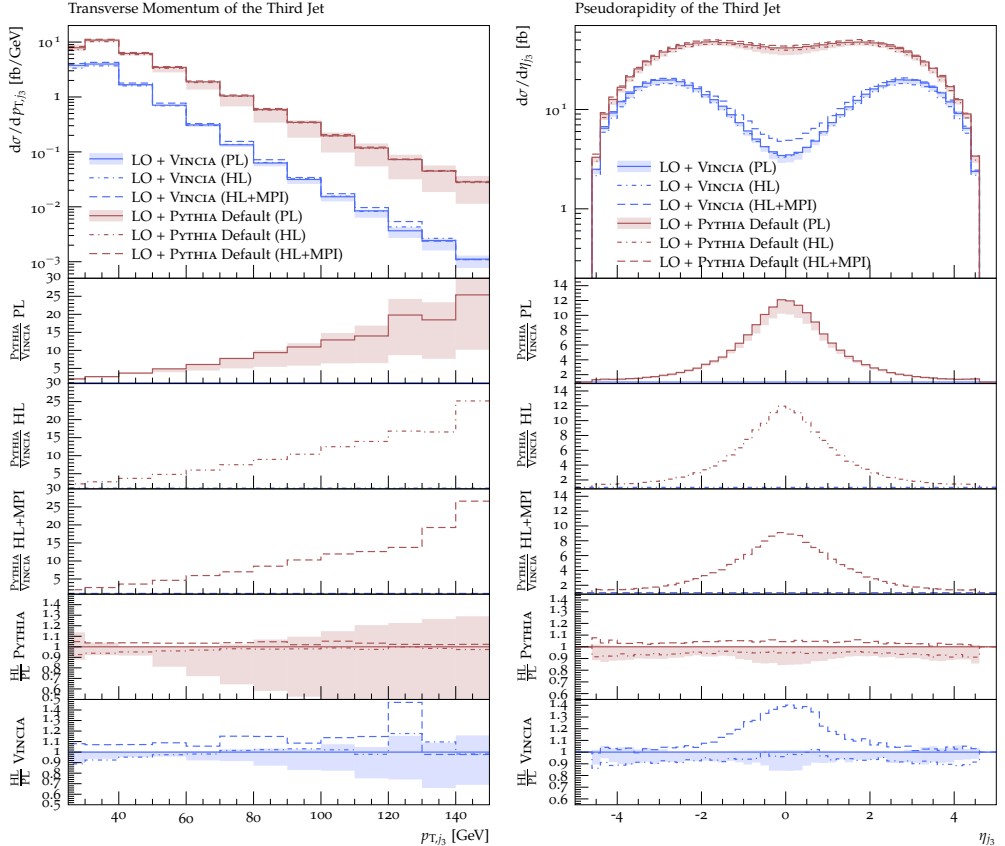

Figure 17: Detailed comparison of PYTHIA DGLAP and VINCIA LO+PS predictions at parton level, hadron level, and hadron-level plus MPI for the transverse momentum (*left*) and pseudorapidity of the third jet (*right*).

PYTHIA's simple shower, and secondly, the discrepancy between the simple shower and VINCIA overpowers the MPI effect greatly. As such, the inclusion of hadron-level and MPI effects emphasise that VINCIA's antenna shower reproduces QCD coherence effects more faithfully than PYTHIA's simple shower.

## 4 Conclusion

We have here studied the effect of QCD radiation in VBF Higgs production, focusing in particular on how the coherent emission patterns exhibited by this process are modelled by various parton-shower approaches that are available in the PYTHIA event generator, and how significant the corrections to that modelling are, from higher fixed-order matrix elements. From a QCD point of view, the main hallmark of VBF is that gluon emission in the central region originates from intrinsically coherent interference between initial- and final-state radiation. In DGLAP-style showers, which are anchored in the collinear limits and treat ISR and FSR separately, this interplay can only be captured at the azimuthally integrated level via angular ordering, while it is a quite natural element in dipole- and antenna-based formalisms, in which initial-final colour flows enter on an equal footing with final-final and initial-initial flows. Hence we would expect the latter (dipole/antenna-style) approaches to offer more robust and reliable modelling of the radiation patterns in VBF than the former (DGLAP-based) approaches.

To this end, we have compared the VINCIA antenna shower to PYTHIA's default ("simple")

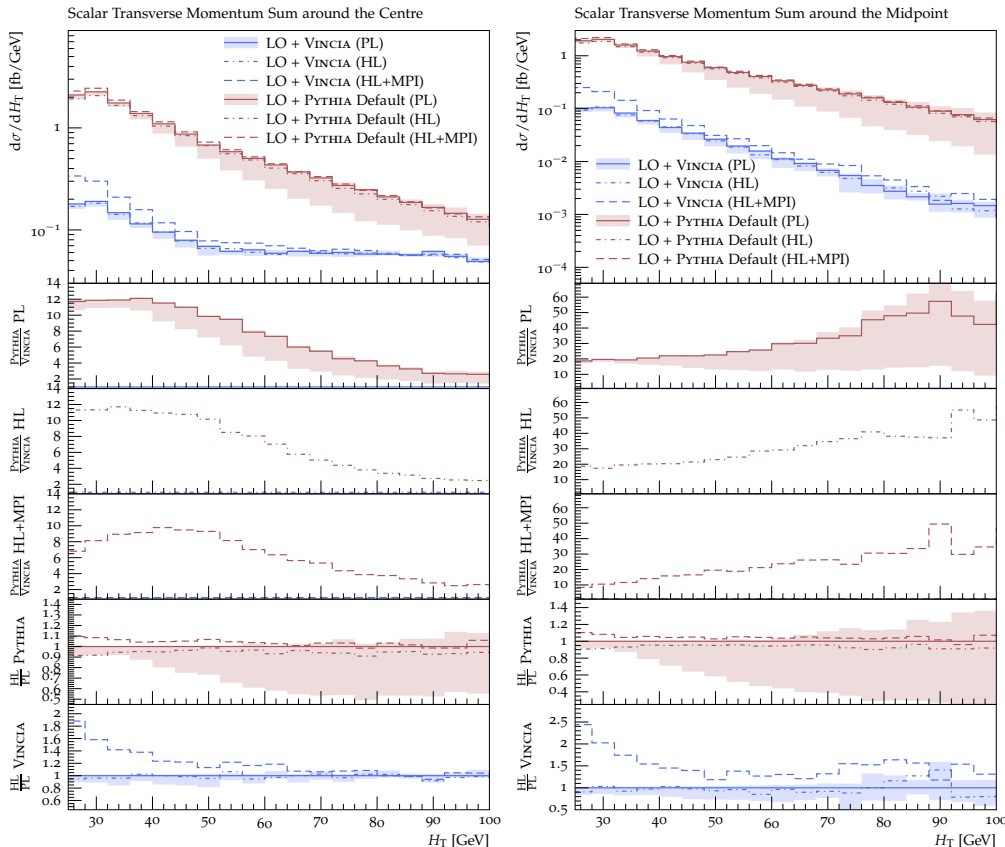

Figure 18: Detailed comparison of PYTHIA DGLAP and VINCIA LO+PS predictions at parton level predictions for the central $H_T$ (*left*) and midpoint $H_T$ (*right*).

shower, including both its (default) DGLAP and its dipole-improved option ("dipole recoil"). We have shown that at leading order, large discrepancies pertaining to the radiation of additional jets in the central rapidity regions exist between the default PYTHIA predictions and the ones obtained with the dipole option and VINCIA, while the latter two appear more consistent. This effect even concerns observables related to the tagging jets, i.e. those jets which are described by the matrix element and not the shower. We have confirmed that these findings apply to both external (LHA) and internal events.

After matching the showers to the NLO, these discrepancies mostly vanish for observables sensitive to the tagging jets or third jet only, while larger effects remain visible in observables sensitive to higher jet multiplicities. These findings are largely consistent with the ones from an earlier study [8], although it is worth highlighting that the disagreement found for the default PYTHIA shower is fairly less pronounced here after matching it to the NLO via the POWHEG scheme. We consider this to be an effect of a more careful treatment of the ordering-variable mismatch between POWHEG and PYTHIA. Based on this, we recommend varying the `POWHEG:pThard` mode contained in the `PowhegHooks` classes to gain an estimate of systematic matching uncertainties. To reduce the uncertainties pertaining to the use of vetoed showers with POWHEG samples, a truncated and vetoed shower should be used with both PYTHIA and VINCIA. As alluded to above, such a scheme is not (yet) available for either of the showers considered in the present study.

In addition to NLO matching, we have studied the effect of including higher-multiplicity tree-level matrix elements in the shower via the CKKW-L merging scheme in VINCIA. We have confirmed that the NLO-matched and one-jet merged calculations lead to comparable predictions for observables sensitive to the third jet. For a set of inclusive observables, we presented

predictions from a tree-level merged calculation at $\mathcal{O}(\alpha_S^4)$. This yields corrections of the order of 20% in the hard tail above around 60 G$e$V of the transverse momentum spectrum of the Higgs-plus-tagging-jet system. Considering the mild corrections in the ranges studied here, it is evident that the sample with four additional jets (i.e. the $2+4$-jet sample) will contribute significantly only in the very hard tails $H_T \gg 100$ G$e$V and $p_{\perp,Hjj} \gg 150$ G$e$V.

Although not the main focus of this study, we have gained a first estimate of non-perturbative corrections on the observables studied here. While we generally found only minor changes from the inclusion of hadron-level corrections, the inclusion of MPIs had a relatively more significant effect on VINCIA's predictions than on the ones obtained with PYTHIA's default shower. This affected the rate of radiation in soft as well as central pseudorapidity regions, i.e., precisely the regions in which VINCIA predicts a strong coherent suppression, so that the MPI contamination becomes relatively more important.

With this study we also proposed two new observables, the scalar transverse momentum sum in the central pseudorapidity region and around the pseudorapidity midpoint between the two tagging jets. We have shown that both of these observables are sensitive to multi-jet radiation, but highlighted that the former becomes dominated by the tagging jets in the hard region $H_T \gtrsim 60$ G$e$V. As an alternative, we demonstrated that the $H_T$ sum around the midpoint between the tagging jets is free of this contamination, with the Born sample only giving a negligible contribution. Due to the strong suppression of radiation in this region, both observables do however receive corrections from the modelling of multi-parton interactions, which would be relevant to study further.

While it has been considered a coherent shower before, this has been the first time that the radiation pattern of the VINCIA antenna shower was studied with a dedicated focus on its coherence. At the same time, we have here showcased NLO matching and tree-level merging methods with VINCIA, which are both publicly available as of the PYTHIA 8.306 release.

# Acknowledgements

We acknowledge support from the Monash eResearch Centre and eSolutions-Research Support Services through the MonARCH HPC Cluster. This work was further partly funded by the Australian Research Council via Discovery Project DP170100708 — "Emergent Phenomena in Quantum Chromodynamics". CTP is supported by the Monash Graduate Scholarship, the Monash International Postgraduate Research Scholarship, and the J.L. William Scholarship. This research was supported by Fermi National Accelerator Laboratory (Fermilab), a U.S. Department of Energy, Office of Science, HEP User Facility. Fermilab is managed by Fermi Research Alliance, LLC (FRA), acting under Contract No. DE-AC02-07CH11359.

# A POWHEG+VINCIA Setup

As mentioned in section 2.3.1, a dedicated vetoed-shower `UserHook` for POWHEG+VINCIA was developed as part of this work and is included in the standard PYTHIA distribution from version 8.306 onwards. At the time of submission of this manuscript, it is included in the file `PowhegHooksVincia.h`, in the directory `include/Pythia8Plugins/`, which also contains the standard `PowhegHooks.h` file. (Note that these two files may be merged into one in a future release; if so, simply omit the corresponding step below.)

Assuming you have a main program that is set up to run POWHEG+PYTHIA (such as the example program `main31.cc` included with PYTHIA), the following changes (highlighted in red) will modify it to run POWHEG+VINCIA:

- Include the `PowhegHooksVincia.h` header file:
  `#include "Pythia8Plugins/PowhegHooksVincia.h"`
  (you can leave any existing `#include "Pythia8Plugins/PowhegHooks.h"` statement; the two will not interfere with each other).

- Replace the POWHEG+PYTHIA user hook pointer by a POWHEG+VINCIA one:
  `shared_ptr<PowhegHooks> powhegHooks;`
  `powhegHooks = make_shared<PowhegHooksVincia>();`
  `pythia.setUserHooksPtr((UserHooksPtr)powhegHooks);`

In addition, the following settings should be used:

- Switch on VINCIA's showers and allow them to fill all of phase space:
  `PartonShowers:model = 2    # Use Vincia's shower algorithm.`
  `Vincia:pTmaxMatch   = 2    # Power showers (to be vetoed by hook).`

- Enable shower vetoes via the `PowhegHooksVincia` (same as for `PowhegHooks`):
  `POWHEG:veto        = 1      # Turn shower vetoes on.`

- Turn QED/EW showers and interleaved resonance decays off:
  `Vincia:ewMode = 0             # Switch off QED/EW showers.`
  `Vincia:interleaveResDec= off # No interleaved resonance decays.`
  While enabling QED showers (`Vincia:ewMode = 1 | 2`) should not pose any problems in the matching, it is not validated (yet). We recommend against using the EW shower (`Vincia:ewMode = 3`) with the POWHEG matching.

- Since POWHEG-BOX event samples come unpolarised, VINCIA's helicity shower should be turned off (the helicity shower needs a polarised Born state):
  `Vincia:helicityShower  = off # Use helicity-averaged antennae.`
  We note that VINCIA offers the possibility to polarise Born configurations using matrix elements provided via interfaces to external generators. We have not studied this in the present work.

- In the POWHEG-specific settings, the number of outgoing particles in the Born process is defined as usual, e.g. =2 for the $2 \rightarrow 2$ example in `main31.cc`, or =3 for the $2 \rightarrow 3$ VBF-type processes studied in this work:
  `POWHEG:nFinal  = 3`
  `# Number of outgoing particles in the Born process.`

- We highly recommend varying the `POWHEG:pThard` mode, for both PYTHIA and VINCIA, to estimate matching systematics. This is how the shaded bands in most of the plots shown in this paper were obtained.
  `POWHEG:pThard  = 2`
  `# Vary (=0,=1,=2) to estimate matching systematics.`

- We also recommend checking all accepted emissions rather than only the first few:
  `POWHEG:vetoCount = 10000`

- The following settings are simply left at their recommended values (the same as for `main31.cmnd`); see the onlin manual section on POWHEG for details:
  `POWHEG:pTemt   = 0`
  `POWHEG:emitted = 0`
  `POWHEG:pTdef   = 1`

- For completeness, (we note that we have anyway turned both MPI and QED showers off in this study):
  ```
  POWHEG:MPIveto = 0
  POWHEG:QEDveto = 2
  ```

The event files generated by POWHEG should be provided in exactly the same way as for PYTHIA+POWHEG. If the POWHEG events were generated in several separate batches, for instance, the resulting files can be read as usual, using PYTHIA's "subruns" functionality:

```
! Powheg Subruns.
Beams:frameType      = 4
Main:numberOfSubruns = 3
!-------------------------------------------------------------------
Main:subrun       = 0
Beams:LHEF        = POWHEG-BOX-V2/VBF_H/run/pwgevents-0001.lhe
!-------------------------------------------------------------------
Main:subrun       = 1
Main:LHEFskipInit = on
Beams:LHEF        = POWHEG-BOX-V2/VBF_H/run/pwgevents-0002.lhe
!-------------------------------------------------------------------
Main:subrun       = 2
Main:LHEFskipInit = on
Beams:LHEF        = POWHEG-BOX-V2/VBF_H/run/pwgevents-0003.lhe
```

## B   VINCIA CKKW-L Setup

Since PYTHIA version 8.304, the release is shipped with VINCIA's own implementation of the CKKW-L merging technique, suitably modified for sector showers.

In the spirit of the last section, let us again assume you have a main program running CKKW-L merging with PYTHIA's default ("simple") shower. (We note that this is a hypothetical setup for the purpose of this study, as the default merging implementation in PYTHIA 8.3 does not handle VBF processes. An algorithmic fix is planned for PYTHIA version 8.307 or later.) The following changes are needed to alter it to run VINCIA's CKKW-L merging instead, with changes again highlighted in red.

- Turn VINCIA and its sector showers on[6]:
  ```
  PartonShowers:model  = 2     # Use Vincia's shower algorithm.
  Vincia:sectorShowers = on    # Turn sector showers on.
  ```

- Disable VINCIA components that are not (yet) handled by the merging:
  ```
  Vincia:ewMode = 0               # Switch off QED/EW showers.
  Vincia:interleaveResDec= off # No interleaved resonance decays.
  Vincia:helicityShower  = off # Use helicity-averaged antennae.
  ```
  These three limitations are intended to be temporary and may be lifted in future updates; users are encouraged to check for changes mentioning VINCIA's merging implementation in the Update History section of PYTHIA's HTML manual in releases from 8.307 onwards.

- Enable the merging machinery and set the merging scale definition (in this study, all event samples were regulated by a $k_T$ cut, so $k_T$-merging is turned on):

---

[6]We note that as of now, sector showers are on per default in VINCIA and this flag is listed here only for completeness.

```
Merging:doMerging   = on      # Turn merging machinery on.
Merging:doKTMerging = on      # Set kT as merging scale.
```

- Set the merging scale to the desired value in GeV (note that the cuts on the event samples should be more inclusive than the ones in the merging!):
```
Merging:TMS     = 20          # Value of the merging scale in GeV.
```

- Replace the `Process` string by one obeying VINCIA's syntax, i.e. encased in curly brackets and with whitespaces between particles, and switch the dedicated VBF treatment on:
```
Merging:process = {p p > h0 j j} # Define the hard process.
Vincia:mergeVBF = on             # Enable merging in VBF systems.
```

- Set the number of additional jets with respect to the Born process (e.g. for the VBF process considered here, the number of *additional* jets is 4, while the *total* number of jets is 6):
```
Merging:nJetMax = 4  # Merge samples with up to 4 additional jets.
```

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
