# Peer review of "A Study of QCD Radiation in VBF Higgs Production with Vincia and Pythia"

_SciPost Physics, doi:SciPost Phys. 12, 010 (2022)_

## Round 1 · Referee Report · Anonymous (Referee 1) · 2021-7-30

Report

The present manuscript describes predictions for Vector Boson Fusion (VBF) processes at hadron colliders, with an emphasis on parton shower effects and the matching to fixed order, and some first studies of the impact of soft QCD effects. All of these aspects have extensively been discussed in the literature or in talks at recent conferences. The originally new content of the manuscript is the inclusion of (leading order) multijet merging, and the use of the Vincia parton shower algorithm, however the title is not indicative of this fact.

Effects due to additional hard jet radiation are in fact worth to consider in VBF processes; this has so far been done at fixed, well separated jet multiplicities, and merging different multiplicities is certainly a needed addition to these studies. As such the manuscript is a welcome addition to the literature, despite the limited insight it provides into how merging algorithms can be applied to the non-trivial structure of jets present at the Born level, and how uncertainties can be estimated reliably.

Before I can recommend this manuscript for publication I need to raise several points of criticism which I like the authors to address:

The present paper makes excessive use of the notion of 'coherent QCD radiation', including in its title. Nowhere could I find any analysis or proof of coherence properties of the shower algorithms the authors consider, nor are aspects discussed, which would be specific to QCD coherence in VBF.

In fact, at several places the paper contains unproven claims about the Vincia shower exhibiting coherence, supported by statements about how the soft radiation pattern is incorporated in the shower kernerls (which, by itself, is at most a necessary condition for a coherent evolution). The statement that the standard Pythia shower would not include the soft radiation pattern in its kernels in this respect needs to be clarified (since the prove or disprove needs to involve kinematic details, on top of the structure of the kernels and possibly the entire evolution algorithm).

The authors need to remove their claims aboutr coherence or provide detailed explanations and proofs of their statements. An appropriate discussion would then also need to account for a significant amount of recent literature focusing on the accuracy of parton shower algorithms and properties of QCD coherence as encoded in dipole-type parton shower algorithms: nothing along these lines is mentioned or cited. On the same note should the title of the paper be changed to reflect its true content.

The paper highlights several comparisons to the Pythia shower without the dipole recoil option, a comparison which has been performed, and understood, extensively in the literature since several years (the study presented in arXiv:1803.07943 should also be cited in this context). These comparisons do not provide any additional insight and should be removed in favour of the more interesting question of how and why the more physical Pythia evolution agrees or differs with the Vincia algorithm. This understanding is needed to properly discuss how and why the matching and merging algorithms differ. The outcome that merging improves the behaviour of the default Pythia shower is not surprising, but the non-dipole option will still stay unphysical -- I do not understand how this should be a significant result of the study worth highlighting in the conclusions.

The discussion of the VBF process itself lacks several details: In what approximations is the process calculated? Are interferences neglected, are Higgs-Strahlungs-type topologies neglected? If not, how are colour flows assigned? What generating cuts are applied to the hard jets -- none, technical ones at very low thresholds (how large?), or significant ones (of which the impact should then be discussed). What happens to the scale definition inside Pythia of the hard quarks are scattered forwardly?

Besides these general comments, which address a multitude of places in the text, a few more minor comments are:

General: Parton showers should generally not be referred to as models, but algorithms. Coherence is not a floating point quantity -- it is a property of a parton shower algorithm, which is either present, or not. Please clarify 'amount of coherent radiation' and similar terms.

Introduction: Authors talk about 'unique' colour flow in VBF. Please confront with general remarks on VBF approximation above. References [8,9] do not include [10] which is later referred to one of the previously cited ones.

'DGLAP shower' should be replaced by a precise definition including: partitioning of soft radiation, recoil scheme, and evolution variable.

Footnote 2: Consider to remove this comment or quantify the term 'unphysical'.

Section 2, discussion on truncated showers: More details are needed here. Are the transverse momentum definitions all equivalent? If they differ, how can the authors be confident that there is no need for truncated showering? References to other work on truncated showers and matching/merging in the literature, e.g. arXiv:0905.3072, are missing.

Caption of Figure 2 (as well as later): be more specific about what 'default shower scale' means. Similarly, the argument of the strong coupling in the shower is a property of its (possible) resummation properties but not directly a renormalization scale in the usual sense. Please clarify the term 'shower renormalization scale' in this respect.

Conclusions: Reproducing the soft radiation pattern is not at all impossible or hard to achieve for an evolution which separates initial and final state radiation, as, for instance, done in a coherent branching algorithm. The remark made in that respect should be clarified: is it referring to the first or all emissions? In what phase space region/hierarchy of emissions?
  • validity: -
  • significance: -
  • originality: -
  • clarity: -
  • formatting: -
  • grammar: -

Author:  Christian Tobias Preuss  on 2021-08-18  [id 1690]

(in reply to Report 1 on 2021-07-30)

First of all, we would like to thank the referee for their very detailed report.

Before addressing the referee’s concerns in detail, we would like to stress that this paper is not meant to provide an in-depth theoretical analysis of QCD coherence in VBF processes, but offers a possible solution to a concrete phenomenological and experimental issue and provides a proof that Vincia can be applied to non-trivial processes in a realistic setup.
We understand this was not clear from the original title and we have followed the referee’s suggestion to change it accordingly to be more descriptive of its content. On the same lines, we have slightly modified the abstract.

We acknowledge that a definite statement about Vincia’s coherence properties is unjustified without an explicit proof. We have hence removed the one occurrence where such a definite statement was made (in the second-to-last paragraph in the introduction) and softened it to an ‘expectation’, backed by an account of the literature on QCD coherence in antenna and dipole showers (including ones in which the parton-shower accuracy is discussed). In all other places, we have checked that only mild statements, on the level of expectations, are made. We would like to point out that an overview of the different showers employed in this study is given at the beginning of section 2.2, including brief discussions of their kinematics. Further detailed discussions can be found in the provided references

We disagree, however, with the referee’s opinion that the presentation of results obtained with Pythia’s default shower option is redundant. Despite the fact that it has been known for a long time in the theoretical/phenomenological community that Pythia’s “dipole recoil” option should be used in the modelling of VBF/VBS processes, it is still rarely adopted by the experimental collaborations, and as long as this continues to be the case we believe it has merit to keep the comparison. In fact, Pythia’s default shower is still actively being used (and will be continued to be used) for these processes by ATLAS and CMS. Already in the original version of the article, we had included four recent experimental studies to highlight this issue. We believe that providing a convincing set of studies like the present one helps to accelerate the process of adopting new and improved algorithms.

In this context, we would like to point out that we use the term “DGLAP shower” only in the context of “Pythia’s (pT-ordered) DGLAP shower”, a term we consider unambiguous in the scope of this paper. A definition (including the ordering variable and recoil scheme) is given in section 2.2, with due reference made to papers discussing these aspects in more detail.

Moreover, we would like to emphasise that it is in fact not a trivial statement that matching improves the predictions obtained with Pythia’s default shower, as visible by the huge variations with the “pThard” scale definition, which has not been shown before. Specifically, the analysis of uncertainties pertaining to the use of vetoed showers in POWHEG matching with Pythia is absent in the available literature and as such is a genuinely new aspect of our study. We are not aware of any publication that discusses this important subtlety in the detail we do (if at all).
On this note, we wish to point out that testing the reliability of predictions obtained with vetoed showers in the POWHEG method and gauging the size of related uncertainties is in fact a core part of our study, as stated at the end of section 2.3.1. Despite the fact that the transverse momentum definitions in POWHEG-BOX, Pythia, and Vincia are not equivalent (as mentioned in section 2.3.1), the choice to use vetoed power showers reflects the strategy employed for POWHEG matching with Pythia. We comment on this in the conclusions and do point out that truncated showers will be relevant in the future for ultimate control of systematic uncertainties. We have added further references on truncated showers, despite their limited relevance for the case at hand.

Another reason to discuss Pythia’s default shower is that in previous (VBF/VBS) literature, there has been some confusion about default Pythia settings, including (wrong!) statements that the global recoil is the default option, which is only true for initial-state radiation. Furthermore, there seems to be significant confusion pertaining to Pythia’s “dipole recoil” option. In contrast to the expectation suggested by its name, this scheme does not simply use a different recoil scheme in the default shower algorithm but replaces the independent DGLAP evolution of IF dipoles by a dipole-antenna like evolution.
All of these points have been highlighted clearly and unambiguously in our paper.

Regarding the description of the hard event generation, we extended section 2.1 by a new paragraph (now the second paragraph), in which we now discuss further details pertaining to the specific modelling of the VBF process in our simulations.
Where relevant, generation cuts were mentioned already in the original version.
The scale assignment in Pythia for both internal and external events is described in section 3.1. Nevertheless, a remark is made now already in the new paragraph in section 2.1.

The referee may feel that parton showers should be referred to as algorithms, not models. However the term "parton-shower model" is standard usage in the field. While the referee is free to argue against this usage in their own publications, we do not think that our use of the term “parton-shower model” should render our manuscript unpublishable.
Another point of language is that in the expression “amount of coherent radiation” the term “amount” refers to the “amount of radiation” not the “amount of coherence”. As such, the phrase “amount of coherent radiation” should be well defined and does not have the implication that coherence is a floating point quantity. However, we would be willing to reword this sentence if need be.
We do, however, agree that the phrase “unique colour flow of the VBF process” is ambiguous and possibly misleading. Hence, we have replaced it by “distinct colour flow of the VBF process at leading order (LO)”.

We thank the referee for pointing out that reference [8] in the citation [8,9] in the introduction was erroneously pointing to the wrong entry. It was supposed to point to what has previously been reference [10]. This issue is fixed now. We note that the previous reference [8] is discussed at a later point in the introduction, now as reference [19].
We also added references to arXiv:1803.07943 as well as arXiv:2102.10991 and ATL-PHYS-PUB-2019-004 to the introduction.

Concerning some of our terminology, we have now clarified the term ‘unphysical’ in footnote 2, pointing out the incoherent IF kinematics which the technical fix in Pythia 8.242 used and have replaced the term “default shower scale” by “default shower starting scale” in the caption of figure 2. The latter has been defined in equation (7) and the sentence preceding it. Further, we added a footnote describing our use of the term “shower renormalisation scale”. We note that it is in line with the standard terminology in Pythia.

Finally, we agree with the referee’s comment that our statement regarding the reproduction of the soft radiation pattern with DGLAP-based parton showers was not clear. We have hence amended this remark and now acknowledge the treatment in angular-ordered showers.

---

## Round 2 · Referee Report · Anonymous (Referee 1) · 2021-10-3

Report

The authors have improved on the clarity of the manuscript and did address almost all of my criticism, which removes several possible misunderstandings. I still feel that the sentences involving "amount of coherent radiation" need clarification.

I also insist on the change to "parton shower model". Parton showers approximate QCD cross sections in extreme kinematic limits. As such they are exact calculations, and one can (at least in principle, though recently extensively demonstrated in the literature) unambiguously check, if a given algorithm provides an accurate approximation, or not.

Ambiguities arise in terms of subleading origin, and amount to a choice of scheme in the approximate calculation. To this extend (as opposed to the genuine neccessity of modeling in the absence of first-principles calculations) parton showers are a quantitative component of an event generator and hence solely deserve the term "algorithm".
  • validity: -
  • significance: -
  • originality: -
  • clarity: -
  • formatting: -
  • grammar: -

Author:  Christian Tobias Preuss  on 2021-10-13  [id 1843]

(in reply to Report 1 on 2021-10-03)

We appreciate that we have addressed most of the referee's criticism with our updated manuscript.

Following the referee's suggestions, we have replaced the phrase "amount of coherent radiation" by "amount of radiation". We believe that this rewording makes it unambiguous that the observable we introduce serves the purpose of measuring the total radiative deposit in the defined pseudorapidity regions. Furthermore, we have replaced all occurences of the phrase "shower model" by "shower algorithm".

---

## Round 2 · List of Changes

1. Changed title and abstract to be more descriptive of its content.
  2. Reworded statements about coherence in Vincia.
  3. Replaced "unique colour flow of the VBF process" by "distinct colour flow of the VBF process at leading order (LO)" in the introduction.
  4. Fixed reference pointing to the wrong entry and added further references on studies of Pythia's default shower in VBF/VBS.
  5. Added a new paragraph to section 2.1 discussing further details of the modelling of the VBF process.
  6. Added remark on the scale treatment in Pythia to section 2.1
  7. Added further references on truncated showers to section 2.3
  8. Added remark on the treatment of soft radiation in angular-ordered showers to the conclusion.
  9. Clarified some terminology.

---

## Round 3 · List of Changes

1. Replaced the term "parton-shower model" by "parton-shower algorithm" throughout.
  2. Replaced the phrase "amount of coherent radiation" in the introduction by "amount of radiation".

---

## Editorial Decision

published